# Learning Latent Graph Structures and their Uncertainty

**Alessandro Manenti** [1]   **Daniele Zambon** [1]   **Cesare Alippi** [1 2]

## Abstract

Graph neural networks use relational information as an inductive bias to enhance prediction performance. Not rarely, task-relevant relations are unknown and graph structure learning approaches have been proposed to learn them from data. Given their latent nature, no graph observations are available to provide a direct training signal to the learnable relations. Therefore, graph topologies are typically learned on the prediction task alongside the other graph neural network parameters. In this paper, we demonstrate that minimizing point-prediction losses does not guarantee proper learning of the latent relational information and its associated uncertainty. Conversely, we prove that suitable loss functions on the stochastic model outputs simultaneously grant solving two tasks: (i) learning the unknown distribution of the latent graph and (ii) achieving optimal predictions of the target variable. Finally, we propose a sampling-based method that solves this joint learning task [1]. Empirical results validate our theoretical claims and demonstrate the effectiveness of the proposed approach.

## 1. Introduction

Relational information processing has provided breakthroughs in the analysis of rich and complex data coming from, e.g., social networks, natural language, and biology. This side information takes various forms, from structuring the data into clusters to defining causal relations and hierarchies, and enables machine learning models to condition their predictions on dependency-related observations. In this context, predictive models take the form $y = f_\psi(x, A)$, where the input-output relation $x \mapsto y$ – modeled by $f_\psi$

and its parameters in $\psi$ – is conditioned on the relational information encoded in variable $A$. Graph Neural Networks (GNNs) (Scarselli et al., 2008) are one example of models of this kind that rely on a graph structure represented as an adjacency matrix $A$ and have been demonstrated successful in a plethora of applications (Fout et al., 2017; Shlomi et al., 2020).

Indeed, relational information is needed to implement such a relational inductive bias and, in some cases, it is provided at the application design phase. However, more frequently, such topological information is not rich enough to address the problem at hand, and – not seldom – it is completely unavailable. Therefore, Graph Structure Learning (GSL) emerges as an approach to learn the graph topology (Kipf et al., 2018; Franceschi et al., 2019; Yu et al., 2021; Fatemi et al., 2021; Zhu et al., 2021; Cini et al., 2023) alongside the predictive model $f_\psi$. This entails formulating a joint learning process that learns the adjacency matrix $A$ – or a parameterization of it – along with the predictor's parameters $\psi$. This is usually achieved by optimizing a loss function on the model output $y$, e.g., a point prediction measure based on the square or the absolute prediction error.

Different sources of uncertainty affect the graph structure learning process, including epistemic uncertainty in the data and variability inherent in the data-generating process. Learning appropriate models of the data-generating process can provide valuable insights into the modeled environment with uncertainty quantification enhancing explainability and interpretability, ultimately enabling more informed decision-making. Examples of applications are found in the study of infection and information spreading, as well as biological systems (Gomez Rodriguez et al., 2013; Lokhov, 2016; Deleu et al., 2022). It follows that a probabilistic framework is appropriate to accurately capture the uncertainty in the learned relations whenever randomness affects the graph topology. Probabilistic approaches have been devised in recent years. For instance, research carried out by Franceschi et al. (2019); Zhang et al. (2019); Elinas et al. (2020); Cini et al. (2023) propose methods that learn a parametric distribution $P_A^\theta$ over the latent graph structure $A$. However, none of them have studied whether these approaches were able to learn a *calibrated* latent distribution $P_A^\theta$ for $A$, properly reflecting the uncertainty associated with the learned topology.

[1]Università della Svizzera italiana, IDSIA, Lugano, Switzerland [2]Politecnico di Milano, Milan, Italy. Correspondence to: Alessandro Manenti <alessandro.manenti@usi.ch>.

*Proceedings of the 42nd International Conference on Machine Learning*, Vancouver, Canada. PMLR 267, 2025. Copyright 2025 by the author(s).

[1]Code available at https://github.com/allemanenti/Learning-Calibrated-Structures

In this paper, we fill this gap by addressing the joint problem of learning a predictive model yielding optimal point-prediction performance of the output $y$ and, contextually, a calibrated distribution for the latent adjacency matrix $A$. In particular, the novel contributions can be summarized as:

1. We demonstrate that models trained to achieve optimal point predictions do *not* guarantee calibration of the adjacency matrix distribution [Section 4].

2. We provide theoretical conditions on the predictive model and loss function that guarantee simultaneous calibration of the latent variable and optimal point predictions [Section 5].

3. We propose a theoretically grounded sampling-based learning method to address the joint learning problem [Section 5].

4. We empirically validate the theoretical developments and claims presented in this paper and show that the proposed approach outperforms existing methods in solving the joint learning task [Section 6].

Finally, we emphasize the significance of our contribution. The inherent latent nature of $A$ presents substantial learning challenges. Real-world applications rarely provide direct observations of the (latent) graph structure, making it impossible to use such data as learning signals for training the graph distribution $P_A^\theta$. This lack of real-world observations not only hampers model training but also complicates empirical evaluation of the learned latent distribution. Consequently, flawed decisions may be derived from learned models. This work addresses these limitations by (i) establishing theoretical guarantees for more robust learning of the latent variable to mitigate the need for evaluation on real data, and (ii) conducting a rigorous empirical analysis on synthetic datasets that provide the – otherwise missing – ground-truth knowledge required for an accurate validation of our claims.

## 2. Related Work

**Graph Structure Learning**  GSL is often employed end-to-end with a predictive model to better solve a downstream task. Examples include applications within graph deep learning methods for static (Jiang et al., 2019; Yu et al., 2021; Kazi et al., 2022) and temporal data (Wu et al., 2019; 2020; Cini et al., 2023; De Felice et al., 2024); a recent review is provided by Zhu et al. (2021). Some approaches from the literature model the latent graph structure as stochastic (Kipf et al., 2018; Franceschi et al., 2019; Elinas et al., 2020; Shang et al., 2021; Cini et al., 2023), mainly as a way to enforce sparsity of the adjacency matrix. To operate on discrete latent random variables, Franceschi et al.

(2019) utilize straight-through gradient estimations, Cini et al. (2023) rely on score-based gradient estimators, while Niepert et al. (2021) design an implicit maximum likelihood estimation strategy. A different line of research is rooted in graph signal processing, where the graph is estimated from a constrained optimization problem and the smoothness assumption of the signals (Kalofolias, 2016; Dong et al., 2016; Mateos et al., 2019; Coutino et al., 2020; Pu et al., 2021). A few works from the Bayesian literature have tackled the task of estimating uncertainties associated with graph edges. The model-based approaches by Lokhov (2016) and Gray et al. (2020) are two examples tackling relevant applications benefiting from uncertainty quantification. Within the deep learning literature, Zhang et al. (2019) propose a Bayesian Neural Network (BNN) modeling the random graph realizations. Differently, Wasserman & Mateos (2024) develop an interpretable BNN designed over graph signal processing principles using unrolled dual proximal gradient iterations. While some results on the output calibration are exhibited, to the best of our knowledge, no guarantee or evidence of calibration of the latent variable is provided, which we study in this paper instead.

**Calibration of the model's output**  Research on model calibration has primarily focused on obtaining accurate and consistent predictions of the statistical properties of the target (random) variables $y$, from which uncertainty estimates on the model's predictions are derived. For discrete outputs, such as in classification tasks, Guo et al. (2017) investigated the calibration of modern deep learning models and proposed temperature scaling as a solution. Other techniques in the same context include Histogram Binning (Zadrozny & Elkan, 2001), Cross Entropy loss with label smoothing (Müller et al., 2019), and Focal Loss (Mukhoti et al., 2020). For continuous output distributions, Laves et al. (2020) proposed $\sigma$ scaling, while Kuleshov et al. (2018) developed a technique inspired by Platt scaling. More recently, conformal prediction techniques (Shafer & Vovk, 2008) have gained popularity for providing confidence intervals in predictions. We stress that within this paper, we are mainly concerned with latent variable calibration, rather than output calibration, although the two are related to each other.

**Deep latent variable models**  Latent variables are extensively used in deep generative modeling (Kingma & Welling, 2013; Rezende et al., 2014), both with continuous and discrete latent variables (Van Den Oord et al., 2017; Bartler et al., 2019). In deep models, latent random variables often lack direct physical meaning, with only the outputs being collected for training. Therefore, studies mainly focused on maximizing the likelihood of the observed outputs in the training set, rather than calibrating the latent distribution. A few works proposed regularization of the latent space to improve stability and accuracy (Xu & Durrett, 2018; Joo et al.,

2020), facilitate smoother transitions in the output when the latent variable is slightly modified (Hadjeres et al., 2017), and apply other techniques aimed at enhancing data generation or improving model performance in general (Connor et al., 2021).

To the best of our knowledge, no prior work has studied the joint learning problem of calibrating the latent graph distribution while achieving optimal point predictions.

## 3. Problem Formulation

Consider a set of $N$ interacting entities and the data-generating process

$$
\begin{cases} A \sim P_A^* \\ y = f^*(x, A) \end{cases} \tag{1}
$$

where $y \in \mathcal{Y}$ is the system output obtained from input $x \in \mathcal{X}$ through function $f^*$ and conditioned on a realization of the latent adjacency matrix $A \in \mathcal{A} \subseteq \{0,1\}^{N \times N}$ drawn from distribution $P_A^*$; input $x$ is assumed to be drawn from any distribution $P_x^*$ and superscript $*$ refers to unknown entities. Each entry of the adjacency matrix $A$ is a binary value encoding the existence of a pairwise relation between two nodes. In the sequel, $x \in \mathcal{X} \subseteq \mathbb{R}^{N \times d_{in}}$ and $y \in \mathcal{Y} \subseteq \mathbb{R}^{N \times d_{out}}$ are stacks of $N$ node-level feature vectors of dimension $d_{in}$ and $d_{out}$, respectively, representing continuous inputs and outputs.

Given a training dataset $\mathcal{D} = \{(x_i, y_i)\}_{i=1}^n$ of $n$ input-output observations from (1), we aim at learning a probabilistic predictive model

$$
\begin{cases} A \sim P_A^\theta \\ \hat{y} = f_\psi(x, A) \end{cases} \tag{2}
$$

from $\mathcal{D}$, while learning at the same time distribution $P_A^\theta$ approximating $P_A^*$. The two parameter vectors $\theta$ and $\psi$ are trained to approximate distinct entities in (1), namely the distribution $P_A^*$ and function $f^*$, respectively. We assume

**Assumption 3.1.** The family $\{P_A^\theta\}$ of probability distributions $P_A^\theta$ parametrized by $\theta$ and the family of predictive functions $\{f_\psi\}$ are expressive enough to contain the true latent distribution $P_A^*$ and function $f^*$, respectively.

Assumption 3.1 implies that $f^* \in \{f_\psi\}$ and $P_A^* \in \{P_A^\theta\}$ but does not request uniqueness of the parameters vectors $\psi^*$ and $\theta^*$ such that $f_{\psi^*} = f^*$ and $P_A^{\theta^*} = P_A^*$. Under such assumption the minimum function approximation error is null and we can focus on the theoretical conditions requested to guarantee successful learning, i.e., achieving both optimal point predictions and latent distribution calibration. In Section 6.2, we empirically show that the theoretical results can extend beyond this assumption in practice.

**Optimal point predictions** Outputs $y$ and $\hat{y}$ of probabilistic model (1) and (2) are random variables following push-forward distributions [2] $P_{y|x}^*$ and $P_{y|x}^{\theta,\psi}$, respectively. A single point prediction $y_{PP} \in \mathcal{Y}$ can be obtained through an appropriate functional $T[\cdot]$ as

$$
y_{PP} = y_{PP}(x, \theta, \psi) \equiv T\left[P_{y|x}^{\theta,\psi}\right]. \tag{3}
$$

For example, $T$ can be the expected value or the value at a specific quantile. We then define an *optimal predictor* as one whose parameters $\theta$ and $\psi$ minimize the expected *point-prediction loss*

$$
\mathcal{L}^{point}(\theta, \psi) = \mathbb{E}_{x \sim P_x^*}\left[\mathbb{E}_{y \sim P_{y|x}^*}\left[\ell\big(y, y_{PP}(x, \theta, \psi)\big)\right]\right] \tag{4}
$$

between the system output $y$ and the point-prediction $y_{PP}$, as measured by of a loss function $\ell : \mathcal{Y} \times \mathcal{Y} \to \mathbb{R}_+$.

Statistical functional $T$ is coupled with the loss $\ell$ as the optimal functional $T$ to employ given a specific loss $\ell$ is often known (Berger, 1990; Gneiting, 2011), when $P_{y|x}^{\theta,\psi}$ approximates well $P_{y|x}^*$. For instance, if $\ell$ is the Mean Absolute Error (MAE) the associated functional $T$ is the median, if $\ell$ is the Mean Squared Error (MSE) the associated functional is the expected value.

**Latent distribution calibration** Calibration of a parametrized distribution $P_A^\theta$ requires learning parameters $\theta$, so that $P_A^\theta$ aligns with true distribution $P_A^*$. Quantitatively, a dissimilarity measure $\Delta^{cal} : \mathcal{P}_A \times \mathcal{P}_A \to \mathbb{R}_+$, defined over a set $\mathcal{P}_A$ of distributions on $\mathcal{A}$, assesses how close two distributions are. The family of $f$-divergences (Rényi, 1961), such as the Kullback-Leibler divergence, and the integral probability metrics (Müller, 1997), such as the maximum mean discrepancy (Gretton et al., 2012) are examples of such dissimilarity measures. In this paper, we are interested in those discrepancies for which $\Delta^{cal}(P_1, P_2) = 0 \iff P_1 = P_2$ holds. It follows that the latent distribution $P_A^\theta$ is *calibrated* on $P_A^*$ if it minimizes the *latent distribution loss*

$$
\mathcal{L}^{cal} = \mathbb{E}_{x \sim P_x^*}\left[\Delta^{cal}\left(P_A^*, P_A^\theta\right)\right], \tag{5}
$$

or simply $\mathcal{L}^{cal} = \Delta^{cal}\left(P_A^*, P_A^\theta\right)$, when $A$ and $x$ are independent.

The problem of designing a predictive model (2) that both yields optimal point predictions (i.e., minimizes $\mathcal{L}^{point}$ in (4)) and calibrates the latent distribution (i.e., minimizes $\mathcal{L}^{cal}$ in (5)) is non-trivial for two main reasons. At first, as the latent distribution $P_A^*$ is unknown (and no samples from it are available), we cannot directly estimate $\mathcal{L}^{cal}$. Second, as shown in Section 4, multiple sets of $\theta$ parameters may minimize $\mathcal{L}^{point}$ without minimizing $\mathcal{L}^{cal}$.

---

[2] The distribution of $y = f^*(x, A)$ originated from $P_A^*$ and of $\hat{y} = f_\psi(x, A)$ originated from $P_A^\theta$.

# 4. Limitations of Point-Prediction Optimization

In this section, we demonstrate that the optimization of a point prediction loss (Equation (4)) does not generally grant calibration of the latent random variable $A$.

**Proposition 4.1.** *Consider Assumption 3.1. Loss function $\mathcal{L}^{point}(\theta, \psi)$ in (4) is minimized by all $\theta$ and $\psi$ s.t. $T\left[P_{y|x}^{\theta,\psi}\right] = T\left[P_{y|x}^*\right]$ almost surely on $x$ and, in particular,*

$$\mathcal{L}^{point}(\theta, \psi) \text{ is minimal} \quad \substack{\Longleftarrow \\ \not\Longrightarrow} \quad P_{y|x}^{\theta,\psi} = P_{y|x}^*.$$

The proof of the proposition is given in Appendix A.1; we provide a counterexample for which calibration is not granted even when the processing function $f_\psi$ is equal to $f^*$ in Appendix A.2.

The limitation of point-prediction losses is also empirically demonstrated in Section 6.3, Table 2, where it is shown that optimizing point-prediction losses does not grant calibration

Given the provided negative result and the impossibility of assessing loss $\mathcal{L}^{cal}$ in (5), in the next section, we propose another optimization objective that, as we will prove, allows us to both calibrate the latent random variable and to have optimal point predictions.

# 5. Predictive Distribution Optimization: Two Birds with One Stone

In this section, we show that we can achieve an optimal point predictor (2) and a calibrated latent distribution $P_A^\theta$ by comparing push-forward distributions $P_{y|x}^*$ and $P_{y|x}^{\theta,\psi}$ of the outputs $y$ conditioned on input $x$. In particular, Theorem 5.2 below proves that, under appropriate conditions, minimization of the *output distribution loss*

$$\mathcal{L}^{dist}(\theta, \psi) = \mathbb{E}_{x \sim P_x^*}\left[\Delta(P_{y|x}^*, P_{y|x}^{\theta,\psi})\right] \quad (6)$$

provides calibrated $P_A^\theta$, even when $P_A^*$ is not available; $\Delta : \mathcal{P}_y \times \mathcal{P}_y \to \mathbb{R}_+$ is a dissimilarity measure between distributions over space $\mathcal{Y}$. We assume the following on dissimilarity measure $\Delta$.

**Assumption 5.1.** $\Delta(P_1, P_2) \geq 0$ for all distributions $P_1$ and $P_2$ in $\mathcal{P}_y$ and $\Delta(P_1, P_2) = 0$ if and only if $P_1 = P_2$.

Several choices of $\Delta$ meet Assumption 5.1, e.g., $f$-divergences and some integral probability metrics (Müller, 1997); the dissimilarity measure $\Delta$ employed in this paper is discussed in Section 5.1.

**Theorem 5.2.** *Let $I = \{x : A \mapsto f^*(x, A) \text{ is injective}\} \subseteq \mathcal{X}$ be the set of points $x \in \mathcal{X}$ such that map $A \mapsto f^*(x, A)$*

*is injective. Under Assumptions 3.1 and 5.1, if $\mathbb{P}_{x \sim P_x^*}(I) > 0$ and $\psi^*$ is such that $f_{\psi^*} = f^*$, then*

$$\mathcal{L}^{dist}(\theta, \psi^*) = 0 \implies \begin{cases} \mathcal{L}^{point}(\theta, \psi^*) \text{ is minimal} \\ \mathcal{L}^{cal}(\theta) = 0. \end{cases}$$

Theorem 5.2 is proven in Appendix A.3. Under the theorem's hypotheses, a predictor that minimizes $\mathcal{L}^{dist}$ is both *calibrated* on the latent random distribution and provides *optimal point predictions*. This overcomes limits of Proposition 4.1 where optimization of $\mathcal{L}^{point}(\theta, \psi^*)$ does not grant $\mathcal{L}^{cal}(\theta) = 0$.

The hypotheses under which Theorem 5.2 holds are rather mild. In fact, condition $\mathbb{P}_{x \sim P_x^*}(I) > 0$ pertains to the data-generating process and intuitively ensures that, for some $x$, different latent random variables produce different outputs. A sufficient condition for $\mathbb{P}_{x \sim P_x^*}(I) > 0$ to hold is the existence of a point $\bar{x}$ in the support of $P_x^*$ such that $A \mapsto f^*(\bar{x}, A)$ is injective with $f^*$ continuous w.r.t. $\bar{x}$; see Corollary A.1 in Appendix A.3. Although only a single point $\bar{x}$ is required, having more points that satisfy the condition simplifies the training of the parameters. Corollary A.1 holds for arbitrarily complex processing functions $f^*$. More specifically, when considering simple GNN layers and discrete latent matrices $A$, we can prove that the condition $\mathbb{P}_{x \sim P_x^*}(I) > 0$ is − except from pathological cases − always satisfied (see Proposition A.2 in Appendix A.3). Instead, condition $f_\psi = f^*$ is set to avoid scenarios of different, yet equivalent, [3] representations of the latent distribution. An empirical analysis of the theorem's assumptions is provided in Section 6.2, demonstrating that the theoretical results hold in practice, even when those assumptions do not strictly apply.

Assumptions 3.1 and 5.1 can be met with an appropriate choice of model (2) and measure $\Delta$; as such they are controllable by the designer. Assumption 5.1 prevents from obtaining mismatched output distributions when $\mathcal{L}^{dist}(\theta, \psi) = 0$ and can be easily satisfied. As mentioned above, popular measures, e.g., the Kullback-Leibler divergence, meet the theorem's assumptions and therefore can be adopted as $\Delta$. However, as $f$-divergences rely on the explicit evaluation of the likelihood of $y$, they are not always practical to compute (Mohamed & Lakshminarayanan, 2016). For this reason, we propose considering the Maximum Mean Discrepancy (MMD) (Gretton et al., 2012) as a versatile alternative that allows Monte Carlo computation without requiring evaluations of the likelihood w.r.t. the output distributions $P_{y|x}^*$ and $P_{y|x}^{\theta,\psi}$. Energy distances (Székely & Rizzo, 2013) provide an alternative feasible choice.

---

[3]E.g., $f_\psi(A, x) = f_*(\mathbf{1} - A, x)$ and $P_A^\theta$ encoding the absence of edges instead of their presence as in $P_A^*$.

## 5.1. Maximum Mean Discrepancy

Given two distributions $P_1, P_2 \in \mathcal{P}_y$, the MMD can be defined as

$$\text{MMD}_{\mathcal{G}}[P_1, P_2] = \sup_{g \in \mathcal{G}} \left\{ \mathbb{E}_{y \sim P_1} \left[ g(y) \right] - \mathbb{E}_{y \sim P_2} \left[ g(y) \right] \right\},$$
(7)

i.e., the supremum, taken over a set $\mathcal{G}$ of functions $\mathcal{Y} \to \mathbb{R}$, of the difference between expected values w.r.t. $P_1$ and $P_2$. An equivalent form is derived for a generic kernel function $\kappa(\cdot, \cdot) : \mathcal{Y} \times \mathcal{Y} \to \mathbb{R}$:

$$\text{MMD}^2_{\mathcal{G}_\kappa}[P_1, P_2] = \mathbb{E}_{y_1, y_1' \sim P_1} \left[ \kappa(y_1, y_1') \right]$$
$$- 2 \mathbb{E}_{\substack{y_1 \sim P_1 \\ y_2 \sim P_2}} \left[ \kappa(y_1, y_2) \right] + \mathbb{E}_{y_2, y_2' \sim P_2} \left[ \kappa(y_2, y_2') \right], \quad (8)$$

and it is associated with the unit-ball $\mathcal{G}_k$ of functions in the reproducing kernel Hilbert space of $\kappa$; note that (8) is the square of (7). Moreover, when universal kernels are considered (e.g., the Gaussian one), then (8) fulfills Assumption 5.1 (see Theorem 5 in (Gretton et al., 2012)). Dissimilarity in (8) can be conveniently estimated via Monte Carlo (MC) and employed within a deep learning framework. Accordingly, we set $\Delta = \text{MMD}^2_{\mathcal{G}_\kappa}$ and learn parameter vectors $\psi$ and $\theta$ by minimizing $\mathcal{L}^{dist}(\theta, \psi)$ via gradient-descent methods.

## 5.2. Finite-Sample Computation of the Loss

To compute the gradient of $\mathcal{L}^{dist}(\theta, \psi) = \mathbb{E}_{x \sim P_x^*} \left[ \text{MMD}^2_{\mathcal{G}_\kappa} \left[ P_{y|x}^{\theta, \psi}, P_{y|x}^* \right] \right]$ w.r.t. parameter vectors $\psi$ and $\theta$, we rely on MC sampling to estimate in (6) expectations over input $x \sim P_x^*$, target output $y \sim P_{y|x}^*$ and model output $\hat{y} \sim P_{y|x}^{\theta, \psi}$. This amounts to substitute $\text{MMD}^2_{\mathcal{G}_\kappa}$ with

$$\widehat{\text{MMD}}^2_{\theta, \psi}(x, y) = 2 \frac{\sum_{j<i=1}^{N_{adj}} \kappa(\hat{y}_i, \hat{y}_j)}{N_{adj}(N_{adj} - 1)} - 2 \frac{\sum_{i=1}^{N_{adj}} \kappa(y, \hat{y}_i)}{N_{adj}}.$$
(9)

In (9), $N_{adj} > 1$ is the number of adjacency matrices sampled from $P_A^\theta$ to obtain output samples $\hat{y}_i = f_\psi(x, A_i) \sim P_{y|x}^{\theta, \psi}$, whereas the pair $(x, y)$ is a pair from the training set $\mathcal{D}$. We remark that in (9) the third term of (8) – i.e., the one associated with the double expectation with respect to $P_{y|x}^*$ – is neglected as it does not depend on $\psi$ and $\theta$.

Gradient $\nabla_\psi \mathcal{L}^{dist}(\theta, \psi)$ is computed via automatic differentiation by averaging $\nabla_\psi \widehat{\text{MMD}}^2(\theta, \psi)$ within a mini-batch of observed data pairs $(x_i, y_i) \in \mathcal{D}$. For $\nabla_\theta \mathcal{L}^{dist}(\theta, \psi)$, the same approach is not feasible. This limitation arises because the gradient is computed with respect to the same parameter vector $\theta$ that defines the integrated distribution. Here, we rely on a score-function gradient estimator (SFE)

(Williams, 1992; Mohamed et al., 2020) which uses the log derivative trick to rewrite the gradient of an expected loss $L$ as $\nabla_\theta \mathbb{E}_{A \sim P^\theta}[L(A)] = \mathbb{E}_{A \sim P^\theta}[L(A) \nabla_\theta \log P^\theta(A)]$, with $P^\theta(A)$ denoting the likelihood of $A \sim P^\theta$. Applying the SFE to our problem the gradient w.r.t. $\theta$ reads:

$$\nabla_\theta \mathcal{L}^{dist} = \mathbb{E}_{x, y^*} \Bigg[$$
$$\mathbb{E}_{\hat{y}_1, \hat{y}_2} \left[ \kappa(\hat{y}_1, \hat{y}_2) \nabla_\theta \log \left( P_{y|x}^{\theta, \psi}(\hat{y}_1) P_{y|x}^{\theta, \psi}(\hat{y}_2) \right) \right]$$
$$- 2 \mathbb{E}_{\hat{y}} \left[ \kappa(y^*, \hat{y}) \nabla_\theta \log P_{y|x}^{\theta, \psi}(\hat{y}) \right] \Bigg] \quad (10)$$

where $\hat{y}_1, \hat{y}_2, \hat{y} \sim P_{y|x}^{\theta, \psi}$. An apparent setback of SFEs is their high variance (Mohamed et al., 2020), which we address in Section 5.3 by deriving a variance-reduction technique based on control variates that requires negligible computational overhead.

## 5.3. Variance-Reduced Loss for SFE

Two natural approaches to reduce the variance of MC estimates of (10) involve (i) increasing the number $B$ of training data points in the mini-batch used for each gradient estimate and (ii) increasing the number $N_{adj}$ of adjacency matrices sampled for each data point in (9). These techniques act on two different sources of noise. Increasing $B$ decreases the variance coming from the data-generating process, whereas increasing $N_{adj}$ improves the approximation of the predictive distribution $P_{y|x}^{\theta, \psi}$. Nonetheless, by fixing $B$ and $N_{adj}$, it is possible to further reduce the latter source of variance by employing the *control variates* method (Mohamed et al., 2020) that, in our case, requires only a negligible computational overhead but sensibly improves the training speed (see Section 6).

Consider the expectation $\mathbb{E}_{A \sim P^\theta}[L(A) \nabla_\theta \log P^\theta(A)]$ of the SFE – both terms in (10) can be cast into that form. With the control variates method, a function with null expectation is subtracted from $L(A) \nabla_\theta \log P^\theta(A)$.

$$G(A) = L(A) \nabla_\theta \log P^\theta(A) - \beta \Big( h(A) - \mathbb{E}_{A \sim P^\theta}[h(A)] \Big)$$
(11)

that leads to a reduced variance in the MC estimator of the gradient while maintaining it unbiased. In this paper, we set function $h(A)$ to $\nabla_\theta \log P^\theta(A)$ and show how to compute a near-optimal choice for scalar value $\beta$, often called *baseline* in the literature. As the expected value of $\nabla_\theta \log P^\theta(A)$ is zero, gradient (10) rewrites as

$$\nabla_\theta \mathcal{L}^{dist} = \mathbb{E}_{x, y^*} \Bigg[ - 2 \mathbb{E}_A \big[ (\kappa(y^*, \hat{y}) - \beta_2) \nabla_\theta \log P_A^\theta(A) \big]$$
$$+ \mathbb{E}_{A_1 A_2} \left[ (\kappa(\hat{y}_1, \hat{y}_2) - \beta_1) \nabla_\theta \log \left( P_A^\theta(A_1) P_A^\theta(A_2) \right) \right] \Bigg].$$
(12)

In Appendix B, we show that in our setup the best values of $\beta_1$ and $\beta_2$ are approximated by

$$\tilde{\beta}_1 = \mathbb{E}_x\Big[\mathbb{E}_{A_1,A_2\sim P_A^\theta}\big[\kappa\big(f_\psi(x,A_1),f_\psi(x,A_2)\big)\big]\Big],$$
$$\tilde{\beta}_2 = \mathbb{E}_{x,y^*}\Big[\mathbb{E}_{A\sim P_A^\theta}\big[\kappa\big(y^*,f_\psi(x,A)\big)\big]\Big], \quad (13)$$

which can be efficiently computed via MC reusing the kernel values already computed for (12).

## 5.4. Computational Complexity

Focusing on the most significant terms, for every data pair $(x,y)$ in the training set, computing the loss $\mathcal{L}^{dist}$ requires $\mathcal{O}(N_{adj}^2)$ kernel evaluations $\kappa(\hat{y}_i,\hat{y}_j)$ in (9), $\mathcal{O}(N_{adj})$ forward passes through the GNN $\hat{y}_i = f_\psi(x,A_i)$ in (9) and $\mathcal{O}(N_{adj})$ likelihood computations $P_A^\theta(A_i)$ in (12). The computation of baselines $\beta_1$ and $\beta_2$ in (13) requires virtually no overhead, as commented in previous Section 5.3. Similarly, computing the loss gradients requires $\mathcal{O}(N_{adj}^2)$ derivatives for what concerns the kernels, $\mathcal{O}(N_{adj})$ gradients $\nabla_\psi \hat{y}_i$ and $\nabla_\theta \log P_A^\theta(A_i)$. We empirically observed that for $N_{adj} \geq 16$, both the latent distribution loss $\mathcal{L}^{cal}$ and the point prediction loss $\mathcal{L}^{point}$ of final models are equivalent for the considered problem. This suggests that $N_{adj}$ is not a critical hyperparameter.

Since we can employ sparse representations of adjacency matrices, the GNN processing costs scale linearly in the number of nodes $N$ for bounded-degree graphs. From our experience, the GNN processing is the most demanding operation and the cost of quadratic components, such as the parameterization of $\theta_{ij}$, do not pose significant overhead.

# 6. Experiments

This section empirically validates the proposed technique and the main claims of the paper. While point predictions can be evaluated on observed input-output pairs $(x,y)$ provided as a test set, assessing latent-variable calibration performance – the discrepancy between $P_A^*$ and the learned $P_A^\theta$ – requires knowledge of the ground-truth latent distribution itself or of observation thereof. Such ground-truth knowledge, however, is not available in real-world datasets, as the latent distribution is indeed unknown. Therefore, to validate the theoretical results, we designed a synthetic dataset that allows us to evaluate different performance metrics on both $y$ and $A$. We remark that the latent distribution is used *only* to assess performance and does not drive the model training in any way.

Section 6.1 demonstrates that the proposed approach can successfully solve the joint learning problem across different graph sizes and highlights the effectiveness of the proposed variance reduction technique. Section 6.2 empirically investigates the generality of the theoretical results

we develop, demonstrating appropriate calibration of the latent distribution even in scenarios where the assumptions of Theorem 5.2 do not hold. Section 6.3 demonstrates that the proposed approach is more effective than existing methods in solving the joint learning task. As a last experiment, in Appendix C.5 we test our approach and show that sensible graph distributions can be learned in real-world settings.

**Dataset and models**  Consider data-generating process (1) with latent distribution $P_A^* = P_A^{\theta^*}$ producing $N$-node adjacency matrices. Random graph $A \sim P_A^*$ is given as a set of independent edges $(i,j)$, for $i,j = 1,\ldots,N$, each of which is sampled with probability $\theta_{i,j}^*$. Function $f_* = f_{\psi^*}$ is a generic GNN with node-level readout, i.e., $f_{\psi^*}(\cdot,A) : \mathbb{R}^{N\times d_{in}} \to \mathbb{R}^{N\times d_{out}}$. The components $\theta^*$ are set to either 0 or 3/4 according to the pattern depicted in Figure 1; additional specifics are detailed in Appendix C. We result in a dataset of $35k$ input-output pairs $(x,y)$, 80% of which are used as training set, 10% as validation set, and the remaining 10% as test set. As predictive model family (2), we follow the same architecture of $f_{\psi^*}$ and $P_A^{\theta^*}$ ensuring that during all the experiments Assumption 3.1 is fulfilled; similar models have been used in the literature (Franceschi et al., 2019; Elinas et al., 2020; Kazi et al., 2022; Cini et al., 2023). In Section 6.2 we test the method's validity beyond this assumption. The model parameters are trained by optimizing the expected squared MMD in (9) with the rational quadratic kernel (Bińkowski et al., 2018).

## 6.1. Graph Structure Learning & Optimal Point Predictions

To test our method's ability to both calibrate the latent distribution and make optimal predictions, we train the model minimizing $\mathcal{L}^{dist}$ as described in Section 5.2.

Figure 2 reports the validation losses during training: MMD loss $\mathcal{L}^{dist}$ as in (6), MAE between the learned parameters $\theta$ and the ground truth $\theta^*$ as $\mathcal{L}^{cal}$ (5), and point-prediction loss $\mathcal{L}^{point}$ as in (4) with $\ell$ being the MSE. The results are averaged over 8 different model initializations and error bars report $\pm 1$ standard deviations from the mean. Results are reported with and without applying the variance reduction (Section 5.3), by training only parameters $\theta$ while freezing $\psi$ to $\psi^*$ (same setting of Theorem 5.2), and by jointly training both $\psi$ and $\theta$.

**Solving the joint learning problem**  Figure 2a shows that the training succeeded and the MMD loss $\mathcal{L}^{dist}$ converged to its minimum value. [4] Having minimized $\mathcal{L}^{dist}$, from Figure 2b we see that also the calibration of latent distri-

---

[4]Numerical estimation shows that the minimum value of $\mathcal{L}^{dist}$ for the given kernel is approximately $-0.088$; note that although the $\text{MMD}^2 \geq 0$, the third term in (8) is dropped from (9).

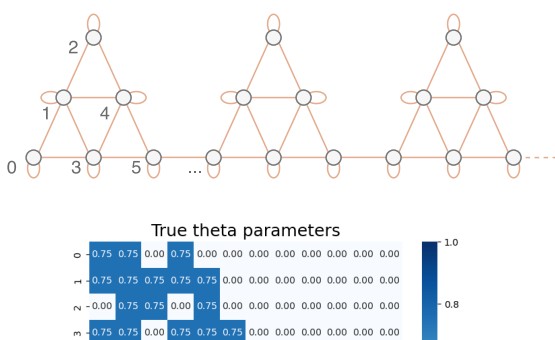

True theta parameters

*Figure 1.* Adjacency matrices sampled from $P_A^* = P_A^{\theta^*}$ for the experiment of Section 6 are subgraphs of the top graph; in this picture, 3 communities of an arbitrarily large graph are shown. Each edge in orange is independently sampled with probability $\theta_{ij}^*$; parameters $\theta_{ij}^*$ defining the edge probabilities are represented at the bottom for a two communities graph.

*Table 1.* Calibration of $P_A^\theta$ under varying levels of misconfiguration for predictive function $f_\psi$. Results are the mean $\pm$ 1 standard deviation assessed over 8 independent runs.

| Max pert. $\Psi$ | MAE on $\theta$ | Max AE on $\theta$ |
|---|---|---|
| 0 | $0.009 \pm 0.001$ | $0.06 \pm 0.01$ |
| 0.1 | $0.010 \pm 0.001$ | $0.07 \pm 0.01$ |
| 0.2 | $0.012 \pm 0.004$ | $0.08 \pm 0.02$ |
| 0.5 | $0.028 \pm 0.011$ | $0.16 \pm 0.06$ |
| 0.8 | $0.047 \pm 0.009$ | $0.28 \pm 0.06$ |

number of free parameters in $\theta$ and the size of the training set can become prohibitive. Nonetheless, in Figure 7, we show all $\sim 15K$ parameters of the considered $P_A^\theta$ can be effectively learned even for relatively large graphs; the final MAE on $\theta$ parameters is 0.003.

### 6.2. Beyond Assumption 3.1

In this section, we empirically study whether Assumption 3.1 is restrictive in practical applications. Specifically, we consider different degrees of model mismatch between the system model in (1) and the approximating model in (2). Unless otherwise specified, we use the same dataset and experimental setup as described in Appendix C.1. Additional details and results are deferred to Appendix C.3.

**Perturbed** $f_{\psi^*}$   As a first experiment, we train $P_A^\theta$ while keeping the parameters of the predictive function $f_\psi$ fixed to a random perturbation of the data-generating model $f^* = f_{\psi^*}$. A perturbed version of $f_\psi^*$ is built by uniformly drawing independent perturbation scalar values $\delta_i \sim \mathcal{U}[-\Psi, \Psi]$, one for each parameter $\psi_i^*$ of $f_{\psi^*}$. Then, each parameter of GNN $f_\psi$ is given as $\psi_i = (1 + \delta_i)\psi_i^*$. Table 1 shows that the learned latent distribution remains reasonably calibrated. Finally, Figures 8-11 show the learned parameter vectors $\theta$ for randomly extracted runs and highlight that the maximum AE of Table 1 is observed only sporadically.

**Generic GNN as** $f_\psi$   In this second experiment, we set $f_\psi$ to be a generic multilayer GNN which we jointly train with graph distribution $P_A^\theta$. The model family $\{f_\psi\}$ employed does not include $f^*$, as $f^*$ uses L-hop adjacency matrices generated from the sampled adjacency matrix $A$, while $f_\psi$ relies on multiple nonlinear 1-hop layers; additional details are reported in Appendix C.3. Upon convergence, models achieved $\mathcal{L}^{point} < 0.19$ using the MSE as loss function $\ell$ in (4); The performance is in line with results in Figure 2c. At last, note that as the GNN used adds self-loops, the diagonal elements of the adjacency matrix are learned as zero, resulting in a larger MAE on $\theta$ (see Figure 12). However, this does not impair learning the off-diagonal $\theta_{ij}$ parameters (i.e., for $i \neq j$). Notably, in the worst-performing model, these off-diagonal parameters achieve a MAE of less than

bution $P_A^\theta$ was successful; in particular, the figure shows that the MAE on $\theta$ parameters ($N^{-2}\|\theta^* - \theta\|_1$) approaches zero as training proceeds (MAE $< 0.01$). Regarding the point predictions, Figure 2c confirms that $\mathcal{L}^{point}$ reached its minimum value; recall that optimal prediction MSE is not 0, as the target variable $y$ is random, and note that a learning rate reduction is applied at epoch number 5. The optimality of the point-prediction is supported also by the performance on separate test data and with respect to the MSE as point-prediction loss $\ell$. Moreover, we observe that calibration is achieved regardless of the variance reduction and whether or not parameters $\psi$ are trained. Lastly, Figure 5 in Appendix C.2 shows the learned parameters $\theta$ of the latent distribution and the corresponding absolute discrepancy resulted from a (randomly chosen) training run.

**Variance reduction effectiveness**   Figures 2a, 2b and 2c demonstrate that the proposed variance reduction method (Section 5.2) yields notable advantages training speed up (roughly 50% faster). For this reason, the next experiments rely on variance reduction.

**Larger graphs**   The theoretical results developed hold for any number of nodes $N$. However, the number of possible edges scales quadratically in the number of nodes – a potential issue inherent to the GSL problem, not our approach. Therefore, for extremely large graphs the ratio between the

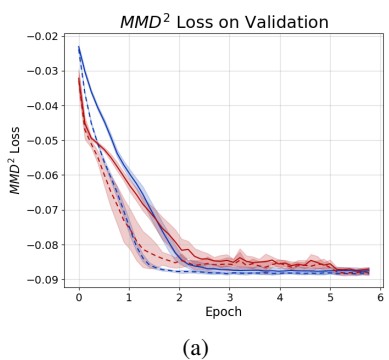 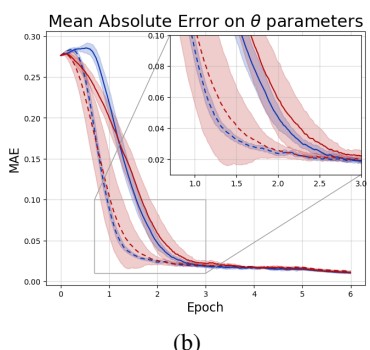 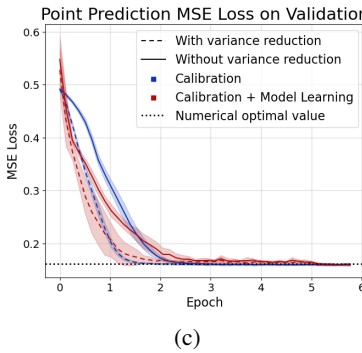

(a)                          (b)                          (c)

*Figure 2.* Validation losses $\mathcal{L}^{dist}$, $\mathcal{L}^{cal}$ and $\mathcal{L}^{point}$ during training. At epoch 5, the learning rate is decreased to ensure convergence. $\mathcal{L}^{dist}$ in Subfigure 2a is negative as the third term in (8) is constant and not considered.

*Table 2.* Calibration and point-prediction performance of models trained by minimizing different loss functions. Losses $\mathcal{L}^{dist}$ follow the approach proposed in this paper. Bold numbers indicate the best-performing models ($p$-value of the Welch's t-test $< 0.01$).

| Train loss | MAE on $\theta$ | MAE on $y$ | MSE on $y$ |
|---|---|---|---|
| $\mathcal{L}^{\text{literature}}_{1,\ell:\,\text{MAE}}$ | $.087 \pm .001$ | $\mathbf{.270 \pm .003}$ | $.180 \pm .003$ |
| $\mathcal{L}^{\text{literature}}_{1,\ell:\,\text{MSE}}$ | $.087 \pm .001$ | $.293 \pm .001$ | $\mathbf{.161 \pm .002}$ |
| $\mathcal{L}^{\text{literature}}_{2,\ell:\,\text{MAE}}$ | $.086 \pm .001$ | $\mathbf{.270 \pm .002}$ | $.176 \pm .002$ |
| $\mathcal{L}^{\text{literature}}_{2,\ell:\,\text{MSE}}$ | $.085 \pm .001$ | $.295 \pm .001$ | $\mathbf{.161 \pm .002}$ |
| $\mathcal{L}^{\text{literature}}_{\text{elbo}}$ | $.082 \pm .001$ | $.310 \pm .010$ | $.191 \pm .020$ |
| $\mathcal{L}^{\text{point}}_{\ell:\,\text{MSE}}$ | $.025 \pm .001$ | $\mathbf{.271 \pm .003}$ | $\mathbf{.161 \pm .002}$ |
| $\mathcal{L}^{\text{dist}}_{\Delta:\,\text{CRPS}}$ | $\mathbf{.010 \pm .002}$ | $\mathbf{.269 \pm .001}$ | $\mathbf{.159 \pm .001}$ |
| $\mathcal{L}^{\text{dist}}_{\Delta:\,\text{MMD}}$ | $\mathbf{.009 \pm .001}$ | $\mathbf{.269 \pm .001}$ | $\mathbf{.159 \pm .001}$ |

0.03, effectively calibrating the latent distribution.

**Misconfigured** $P_A^\theta$    Finally, we violate Assumption 3.1 by fixing $f_\psi = f^*$ and constraining some components of $\theta$ to incorrect values. Specifically, we force parameters $\theta_{i,j}$ for all edges $i, j$ associated with nodes with id 2 and 3 in Figure 1 to 0.25, instead of the correct value of $\theta^*_{i,j} = 0.75$ as in $P_A^*$. Results indicate that the free parameters in $\theta$ are learned appropriately. Notably, increased uncertainty is observed for spurious edges linking to nodes in the first node community (see Figure 1). This is expected given that nearly 60% of the edges in the community were significantly downsampled. Figures 13 and 14 in Appendix C.3 show the learned parameters from randomly selected runs.

### 6.3. Comparison of Loss Functions

As a final experiment, we empirically demonstrate that the proposed choice of loss functions (6) is more effective at calibrating the latent graph distribution, while maintaining or sometimes improving point prediction performance compared to other commonly used loss functions.

**Considered loss functions**    Following our approach we consider two distributional losses, based on the MMD $\mathcal{L}^{\text{dist}}_{\Delta:\,\text{MMD}}$ and the energy distance [5] $\mathcal{L}^{\text{dist}}_{\Delta:\,\text{CRPS}}$. As point prediction loss, we use $\mathcal{L}^{\text{point}}_{\ell:\,\text{MSE}}$ defined in (4) based on the mean squared error. Additionally, we consider three families of losses used in the GSL literature. The first one, defined as

$$\mathcal{L}^{\text{literature}}_{1,\ell} = \mathbb{E}_{x,y^*}\mathbb{E}_{A \sim P_A^\theta}\left[\ell(f_\psi(x, A), y^*)\right] \qquad (14)$$

has been employed, e.g., in Franceschi et al. (2019) and Cini et al. (2023). Note that, differently from $\mathcal{L}^{\text{point}}_{\ell:\text{MSE}}$, the expectation over $A$ is taken outside function $\ell$. The second family, denoted as $\mathcal{L}^{\text{literature}}_{2,\ell}$, is inspired by Kazi et al. (2022). $\mathcal{L}^{\text{literature}}_{2,\ell}$ refines $\mathcal{L}^{\text{literature}}_{1,\ell}$ focusing its optimization to node-level predictions; further details follow in Appendix C.4. For $\mathcal{L}^{\text{literature}}_{1,\ell}$ and $\mathcal{L}^{\text{literature}}_{2,\ell}$, we use both MAE and MSE as $\ell$.

Finally, we adapt the loss function used in (Elinas et al., 2020) for the synthetic regression task:

$$\mathcal{L}^{\text{literature}}_{\text{elbo}} = -\mathbb{E}_{x,y^*}\mathbb{E}_{A \sim P_A^\theta}\left[\log(P^\psi_{y|x^*,A}(y^*))\right] \\ + KL\left[P_A^\theta(A)||\bar{P}_A(A)\right] \quad (15)$$

where $\bar{P}_A(A)$ is a prior distribution and $P^\psi_{y|x^*,A}(y^*)$ is a Gaussian distribution whose mean vector is determined for each node by the GNN output and standard deviation is set as a hyperparameter. We explored different standard deviations and choices of the prior. Details can be found in Appendix C.4

**Results on point prediction**    Table 2 shows that models trained with $\mathcal{L}^{\text{literature}}_{1,\ell}$, $\mathcal{L}^{\text{literature}}_{2,\ell}$ and $\mathcal{L}^{\text{point}}_\ell$ achieve near-optimal [6] point predictions according to their respective

---

[5]By following Section 5.2, the energy distance reduces to the well-known Continuous Ranked Probability Score (CRPS) (Gneiting & Raftery, 2007).

[6]Numerical estimates suggest that the ground-truth optimal MAE and MSE achievable by a predictor are approximately 0.267 and 0.158, respectively.

performance metric (MAE or MSE). Namely, optimizing $\mathcal{L}_{1,\ell:\text{MAE}}^{\text{literature}}$ and $\mathcal{L}_{2,\ell:\text{MAE}}^{\text{literature}}$ leads to minimal MAE, but not to minimal MSE; similarly, optimizing $\mathcal{L}_{1,\ell:\text{MSE}}^{\text{literature}}$ and $\mathcal{L}_{2,\ell:\text{MSE}}^{\text{literature}}$ results in minimal MSE. Conversely, predictors trained with either $\mathcal{L}_{\Delta:\text{MMD}}^{\text{dist}}$ or $\mathcal{L}_{\Delta:\text{CRPS}}^{\text{dist}}$ achieve optimal prediction performance for both metrics. Interestingly, also $\mathcal{L}_{\ell:\text{MSE}}^{\text{point}}$ leads to near-optimal predictions in terms of both MAE and MSE. We attribute the superiority of $\mathcal{L}^{\text{point}}$ over $\mathcal{L}_{1,\ell}^{\text{literature}}$ and $\mathcal{L}_{2,\ell}^{\text{literature}}$ to the use of functional $T$ in (3) which enables a more accurate probabilistic modeling of the data-generating process. A similar observation holds for the calibration error, discussed in the next paragraph.

**Results on calibration**  Optimizing the proposed losses ($\mathcal{L}_{\Delta:\text{MMD}}^{\text{dist}}$ or $\mathcal{L}_{\Delta:\text{CRPS}}^{\text{dist}}$) yields the smallest calibration errors, as measured by the MAE of the latent distribution parameters $\theta$ in $P_A^\theta$. In contrast, loss functions commonly used in the literature result in statistically worse calibration performance. Notably, while the point-prediction loss $\mathcal{L}_{\ell:\text{MSE}}^{\text{point}}$ outperforms $\mathcal{L}_1^{\text{literature}}$ and $\mathcal{L}_2^{\text{literature}}$ in terms of calibration error, it remains statistically inferior to the proposed distributional loss $\mathcal{L}_\Delta^{\text{dist}}$.

We conclude that the proposed approach was the only one capable of effectively solving the joint learning problem.

## 7. Conclusions

Graph structure learning has emerged as a research field focused on learning graph topologies in support of solving downstream predictive tasks. Assuming stochastic latent graph structures, we are led to a joint optimization objective: (i) accurately learning the distribution of the latent topology while (ii) achieving optimal prediction performance on the downstream task. In this paper, at first, we prove both positive and negative theoretical results to demonstrate that appropriate loss functions must be chosen to solve this joint learning problem. Second, we propose a sampling-based learning method that does not require the computation of the predictive likelihood. Our empirical results demonstrate that this approach achieves optimal point predictions on the considered downstream task while also yielding calibrated latent graph distributions.

## Impact Statement

This paper presents work whose goal is to advance the field of Machine Learning. There are many potential societal consequences of our work, none which we feel must be specifically highlighted here.

## Acknowledgments

This work was supported by the Swiss National Science Foundation project FNS 204061: *HORD GNN: Higher-Order Relations and Dynamics in Graph Neural Networks* and partly supported by International Partnership Program of the Chinese Academy of Sciences under Grant 104GJHZ2022013GC.

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

# A. Proofs of the Theoretical Results

## A.1. Minimizing $\mathcal{L}^{point}$ does not guarantee calibration

In this section, we prove Proposition 4.1.

*Proof of Proposition 4.1.* Recall the definition of $\mathcal{L}^{point}$ in (4) using (3)

$$\mathcal{L}^{point}(\psi, \theta) = \mathbb{E}_x\Big[\mathbb{E}_{y^* \sim P^*_{y|x}}\big[\ell\big(y^*, T[P^{\theta,\psi}_{y|x}]\big)\big]\Big]$$

Given loss function $\ell$, $T$ is, by definition (Berger, 1990; Gneiting, 2011), the functional that minimizes

$$\mathbb{E}_{y^* \sim P^*_{y|x}}\Big[\ell\big(y^*, T[P^*_{y|x}]\big)\Big]$$

Therefore, if $P^{\theta,\psi}_{y|x} = P^*_{y|x} \implies \mathcal{L}^{point}$ is minimal. If another distribution over $y$, namely, $P^{\psi',\theta'}_{y|x}$ parametrized by $\theta'$ and $\psi'$ satisfies:

$$T\Big[P^{\psi',\theta'}_{y|x}\Big] = T\Big[P^*_{y|x}\Big]$$

almost surely on $x$, then,

$$\begin{aligned}\mathcal{L}^{point}(\theta', \psi') &= \mathbb{E}_x\Big[\mathbb{E}_{y^* \sim P^*_{y|x}}\big[\ell\big(y^*, T[P^{\psi',\theta'}_{y|x}]\big)\big]\Big] \\ &= \mathbb{E}_x\Big[\mathbb{E}_{y^* \sim P^*_{y|x}}\big[\ell\big(y^*, T[P^*_{y|x}]\big)\big]\Big]\end{aligned}$$

Thus, $P^{\psi',\theta'}_{y|x}$ minimizes $\mathcal{L}^{point}$.

Appendix A.2 discusses graph distributions as counterexamples where $T\big[P^{\psi',\theta'}_{y|x}\big] = T\big[P^*_{y|x}\big]$ but $P^{\psi',\theta'}_{y|x} \neq P^*_{y|x}$. By this, we conclude that reaching the minimum of $\mathcal{L}^{point}(\psi, \theta)$ does not always imply $P^{\psi,\theta}_{y|x} = P^*_{y|x}$. $\qquad\square$

## A.2. Minimizing $\mathcal{L}^{point}$ does not guarantee calibration: an example with MAE

This section shows that $\mathcal{L}^{point}$ equipped with MAE as $\ell$ admits multiple global minima for different parameters $\theta$, even for simple models and $f_\psi = f^*$.

Consider a single Bernoulli of parameter $\theta^* > 1/2$ as latent variable $A$ and a scalar function $f^*(x, A)$ such that $f^*(x, 1) > f^*(x, 0)$ for all $x$. Given input $x$ the value of functional $T(P^*_{y|x})$ that minimizes

$$\mathbb{E}_{y \sim P^*_{y|x}}\Big[\big|y - T[P^*_{y|x}]\big|\Big] = \theta^*\Big|f^*(x, 1) - T[P^*_{y|x}]\Big| + (1 - \theta^*)\Big|f^*(x, 0) - T[P^*_{y|x}]\Big|$$

is $T(P^*_{y|x}) = f^*(x, 1)$; this derives from the fact that range of $f^*$ is $\{f^*(x, 0), f^*(x, 1)\}$ and the likelihood of $f^*(x, 1)$ is larger than that of $f^*(x, 0)$.

Note that $T\big[P^*_{y|x}\big] = f^*(x, 1)$ for all $x$, therefore also $\mathcal{L}^{point}$ is minimized by such $T$. Moreover, $T\big[P^*_{y|x}\big]$ is function of $\theta^*$ and equal to $f^*(x, 1)$ for all $\theta > 1/2$. We conclude that for any $\theta \neq \theta^*$ distributions $P^{\theta,\psi}_{y|x}$ and $P^*_{y|x}$ are different, yet both of them minimize $\mathcal{L}^{point}$ if $\theta > 1/2$.

A similar reasoning applies for $\theta^* < 1/2$.

## A.3. Minimizing $\mathcal{L}^{dist}$ guarantees calibration and optimal point predictions

This section proves Theorem 5.2 and a corollary of it.

*Proof of Theorem 5.2.* Recall from Equation (6) that

$$\mathcal{L}^{dist}(\theta) = \mathbb{E}_x\Big[\Delta(P^*_{y|x}, P^{\theta,\psi}_{y|x})\Big]$$

We start by proving that if $\mathcal{L}^{dist}(\theta, \psi) = 0 \implies \mathcal{L}^{point}(\theta, \psi)$ is minimal.

Note that $\mathcal{L}^{dist}(\theta, \psi) = 0$ implies that $\Delta(P^*_{y|x}, P^{\theta,\psi}_{y|x}) = 0$ almost surely in $x$. Then, by Assumption 5.1, $P^*_{y|x} = P^{\theta,\psi}_{y|x}$ almost surely on $x$ and, in particular, $T[P^*_{y|x}] = T[P^{\theta,\psi}_{y|x}]$, which leads to $\mathcal{L}^{point}(\psi, \theta)$ being minimal (Proposition 4.1).

We now prove that if $\mathcal{L}^{dist}(\theta, \psi^*) = 0 \implies \mathcal{L}^{cal}(\theta) = 0$.

From the previous step, we have that $\mathcal{L}^{dist}(\theta, \psi) = 0$ implies $P^*_{y|x} = P^{\theta,\psi}_{y|x}$ almost surely for $x \in I$. Under the assumption that $f_\psi = f_*$ and the injectivity of $f_*$ in such $x \in I$, for any output $y$ a single $A$ exists such that $f_*(x, A) = y$. Therefore, the probability mass function of $y$ equals that of $A$. Accordingly, $P^*_{y|x} = P^{\theta,\psi}_{y|x}$ implies $P^*_A = P^\theta_A$.

$\square$

We also prove a corollary of Theorem 5.2.

**Corollary A.1.** *Under Assumptions 3.1 and 5.1, if*

1. *$\exists \bar{x} \in Supp(P^*_x) \subseteq \mathcal{X}$ such that $f^*(\bar{x}; \cdot)$ is injective,*

2. *$f^*(x, A)$ is continuous in $x$, $\forall A \in \mathcal{A}$,*

*then*

$$\mathcal{L}^{dist}(\theta, \psi^*) = 0 \implies \begin{cases} \mathcal{L}^{point}(\theta, \psi^*) \text{ is minimal} \\ \mathcal{L}^{cal}(\theta) = 0. \end{cases}$$

The corollary shows that it is sufficient that $f^*$ is continuous in $x$ and there exists one point $\bar{x}$ where $f^*(\bar{x}, \cdot)$ is injective to meet theorem's hypothesis $\mathbb{P}_{x \sim P^*_x}(I) > 0$; we observe that, as $\mathcal{A}$ is discrete, the injectivity assumption is not as restrictive as if the domain were continuous.

*Proof.* As $\mathcal{A}$ is a finite set, the minimum $\bar{\epsilon} = \min_{A \neq A' \in \mathcal{A}} \|f^*(\bar{x}, A) - f^*(\bar{x}, A')\| > 0$ exists and, by the injectivity assumption, is strictly positive.

By continuity of $f^*(\cdot, A)$, for every $\epsilon < \frac{1}{2}\bar{\epsilon}$ there exists $\delta$, such that for all $x \in B(\bar{x}, \delta)$ we have $\|f^*(\bar{x}, A) - f^*(x, A)\| < \epsilon$. It follows that, $\forall x \in B$,

$$\|f^*(x, A) - f^*(x, A')\|$$
$$\geq \|f^*(\bar{x}, A) - f^*(\bar{x}, A')\| - \|f^*(\bar{x}, A) - f^*(x, A)\| - \|f^*(\bar{x}, A') - f^*(x, A')\|$$
$$\geq \|f^*(\bar{x}, A) - f^*(\bar{x}, A')\| - 2\epsilon$$
$$\geq \|f^*(\bar{x}, A) - f^*(\bar{x}, A')\| - \bar{\epsilon} > 0.$$

Where the second inequality holds for the continuity of $x \mapsto f^*(x, A)$ $\forall A$. Finally, as $\bar{x} \in Supp(P^*_x)$ and $B(\bar{x}, \delta) \subseteq I$, we conclude that

$$\mathbb{P}_x(I) \geq \mathbb{P}_x(B(\bar{x}, \delta)) > 0,$$

therefore, we are in the hypothesis of Theorem 5.2 and can conclude that

$$\mathcal{L}^{dist}(\theta, \psi^*) = 0 \implies \begin{cases} \mathcal{L}^{point}(\theta, \psi^*) \text{ is minimal} \\ \mathcal{L}^{cal}(\theta) = 0. \end{cases}$$

$\square$

**A.4. Injectivity hypothesis for graph neural networks**

Now, we show that hypothesis $\mathbb{P}_{x \sim P^*_x}(I) > 0$ of Theorem 5.2 is always met for certain families of graph neural networks.

**Proposition A.2.** *Consider a 1-layer GNN of the form $f^*(x, A) : \sigma(Ax) = y$, with $x, y \in \mathbb{R}^N$ and nonlinear bijective activation function $\sigma$. If the support $Supp(P^*_x)$ of $x$ contains any ball $B$ in $\mathbb{R}^N$ then $\mathbb{P}_{x \sim P^*_x}(I) > 0$.*

To prove Proposition A.2, we rely on following lemma.

**Lemma A.3.** *Given $g(x,a) = ax$ with $a \in \{0,1\}^{1 \times N}$ and $x \in \mathbb{R}^{N \times 1}$. Let $I_g = \{x : g(x,a) \text{ is injective in } a\} \subseteq \mathcal{X}$ be the set of points $x \in \mathcal{X}$ such that map $a \mapsto g(x,a)$ is injective. The following implication holds:*

$$x \notin I_g \iff \exists \delta \neq \mathbf{0} \in \{-1,0,1\}^{1 \times N} \text{ s.t. } \delta \perp x. \tag{16}$$

*Proof.* We prove the two implications separately.

( $\implies$ ) If $x \notin I_g$, then there exist $a', a'' \in \{0,1\}^{1 \times N}$ with $a' \neq a''$ such that $a'x = a''x$. This implies that $(a' - a'')x = 0$. Defining $\delta$ as $(a' - a'')$, we have proven that there exist $\delta \neq \mathbf{0} \in \{-1,0,1\}^{1 \times N}$ such that $\delta x = 0$, i.e., $\delta \perp x$.

( $\impliedby$ ) Assume that $\exists \, \delta \neq \mathbf{0} \in \{-1,0,1\}^{1 \times N}$ such that $\delta \perp x$. Each component $\delta_i$ of $\delta$ can be written as the difference between two values $a'_i, a''_i \in \{0,1\}$. As $\delta \neq \mathbf{0}$ then there exists at least one index $j \in \{1, \ldots, N\}$ such that $a'_j \neq a''_j$. This implies that $\exists \, a', a'' \in \{0,1\}^{1 \times N}$ with $a' \neq a''$ s.t. $(a' - a'')x = 0$, which implies that $x \notin I_g$.

$\square$

*Proof of Proposition A.2.* We begin by considering the projection $\bar{g}(x,a) = ax$ with $a \in \{0,1\}^{1 \times N}$ and $x \in \mathbb{R}^N$. Then we extend to $A \in \{0,1\}^{N \times N}$ and to nonlinear functions.

Let $I_{\bar{g}}^C = \mathbb{R}^N \setminus I_{\bar{g}}$ be the complement in $\mathbb{R}^N$ of $I_{\bar{g}}$. Recalling Lemma A.3 and its notation, we have $3^N - 1$ possible $\delta$, defining a collection of $(3^N - 1)/2$ hyperplanes of vectors $x$ perpendicular to at least one $\delta$; set $I_{\bar{g}}^C$ is the union of such a finite collection of hyperplanes. By hypothesis, $\text{Supp}(P_x^*)$ contains a ball $B \in \mathbb{R}^N$, therefore $\text{Supp}(P_x^*) \not\subset I_{\bar{g}}^C$ and $\mathbb{P}_{x \sim P_x^*}(I_{\bar{g}}^C) < 1$. We conclude that $\mathbb{P}_{x \sim P_x^*}(I_{\bar{g}}) = 1 - \mathbb{P}_{x \sim P_x^*}(I_{\bar{g}}^C) > 0$.

A similar result is proven for $\bar{G}(x,A) = Ax$ with $A \in \{0,1\}^{N \times N}$. In fact, $\bar{G}$ is a stack of $N$ functions $\bar{g}$ above and $I_{\bar{G}} = I_{\bar{g}}$. Finally, composing injective function $G$ with injective function $\sigma$ leads to function $g(x,A) = \sigma(G(x,A))$ being injective in $A$ for the same points $x$ for which $G$ is injective, thus proving the proposition. $\square$

## B. Estimation of Optimal $\beta_1$ and $\beta_2$

Here we show that, when reducing the variance of the SFE via control variates in (12), the best $\beta_1$ and $\beta_2$ can be approximated by

$$\tilde{\beta}_1 = \mathop{\mathbb{E}}_{\substack{x \sim P_x^* \\ A_1, A_2 \sim P_A^\theta}} \Big[ \kappa\left(f_\psi(x, A_1), f_\psi(x, A_2)\right) \Big], \qquad \tilde{\beta}_2 = \mathop{\mathbb{E}}_{\substack{(x,y^*) \sim P_{x,y}^* \\ A \sim P_A^\theta}} \Big[ \kappa\left(y^*, f_\psi(x, A)\right) \Big], \tag{17}$$

Consider generic function $L(A)$ depending on a sample $A$ of a parametric distribution $P_A^\theta(A)$ and the surrogate loss $\tilde{L}(A)$ in (11), i.e.,

$$G(A) = L(A)\nabla_\theta \log P^\theta(A) - \beta\Big(h(A) - \mathbb{E}_{A \sim P^\theta}[h(A)]\Big); \tag{18}$$

Following existing literature (Sutton et al., 1999; Mnih et al., 2016) where $\beta$ is often referred to as *baseline* we set $h(A) = \nabla_\theta \log P^\theta(A)$. The 1-sample MC approximation of the gradient becomes

$$\nabla_\theta \mathbb{E}_{A \sim P^\theta}[L(A)] \approx G(A') = (L(A') - \beta)\nabla_\theta \log P^\theta(A'), \tag{19}$$

with $A'$ sampled from $P_A^\theta$. The variance of the estimator is

$$\mathbb{V}_{A \sim P^\theta}\Big[(L(A) - \beta)\nabla_\theta \log P^\theta(A)\Big] = \mathbb{V}_{A \sim P^\theta}\Big[L(A)\nabla_\theta \log P^\theta(A)\Big] + \\ + \beta^2 \, \mathbb{E}_{A \sim P^\theta}\Big[\left(\nabla_\theta \log P^\theta(A)\right)^2\Big] - 2\beta \, \mathbb{E}_{A \sim P^\theta}\Big[L(A)\left(\nabla_\theta \log P^\theta(A)\right)^2\Big] \tag{20}$$

and the optimal value $\beta$ that minimizes it is

$$\tilde{\beta} = \frac{\mathbb{E}_{A \sim P^\theta}\Big[L(A)\left(\nabla_\theta \log P^\theta(A)\right)^2\Big]}{\mathbb{E}_{A \sim P^\theta}\Big[\left(\nabla_\theta \log P^\theta(A)\right)^2\Big]}. \tag{21}$$

If we approximate the numerator with $\mathbb{E}[L(A)]\mathbb{E}[(\nabla_\theta \log P^\theta(A))^2]$, we obtain that $\tilde{\beta} \approx \mathbb{E}[L(A)]$. By substituting $L(A)$ with the two terms of (10) we get the values of $\beta_1$ and $\beta_2$ in (17).

We experimentally validate the effectiveness of this choice of $\beta$ in Section 6.

## C. Further Experimental Details

### C.1. Dataset description and models

In this section, we describe the considered synthetic dataset, generated from the system model (1). The latent graph distribution $P_A^*$ is a multivariate Bernoulli distribution of parameters $\theta_{ij}^*$: $P_A^* \equiv P_{\theta^*}(A) = \prod_{ij} \theta_{ij}^{*A_{ij}} (1 - \theta_{ij}^*)^{(1-A_{ij})}$. The components of $\theta^*$ are all null, except for the edges of the graph depicted in Figure 3 which are set to $3/4$. A heatmap of the adjacency matrix can be found in Figure 4.

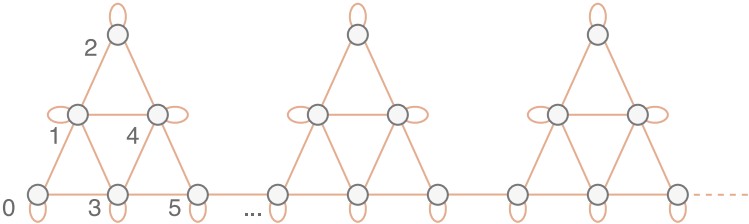

*Figure 3.* The adjacency matrices used in this paper are sampled from this graph. Each edge in orange is independently sampled with probability $\theta^*$. In the picture, 3 communities of an arbitrarily large graph are shown.

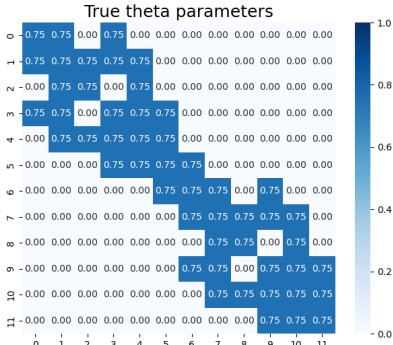

*Figure 4.* $\theta_{ij}^*$ parameters for each edge of the latent adjacency matrix. Each square corresponds to an edge, and the number inside is the probability of sampling that edge for each prediction.

Regarding the GNN function $f^*$, we use the following system model:

$$\begin{cases} y = f_{\psi^*}(A, x) = \tanh\left(\sum_{l=1}^{L} \mathbb{1}[A^l \neq 0]x\psi_l^*\right) \\ A \sim P_{\theta^*}(A) \end{cases} \quad (22)$$

where $\mathbb{1}[\cdot]$ is the element-wise indicator function: $\mathbb{1}[a] = 1 \iff a$ is true. $x \in \mathbb{R}^{N \times d_{in}}$ are randomly generated inputs: $x \sim \mathcal{N}(0, \sigma_x^2 \mathbb{I})$. $\psi_l^* \in \mathbb{R}^{d_{out} \times d_{in}}$ are part of the system model parameters. We summarize the parameters considered in our experiment in Table 3.

*Table 3.* Table of the parameters used to generate the synthetic dataset.

| Parameter | Values |
|---|---|
| $\theta^*$ | 0.75 |
| $\sigma_x$ | 1.5 |
| $N$ | 12 |
| $d_{in}$ | 4 |
| $d_{out}$ | 1 |
| $\psi_1^*$ | $[0.3, -0.2, 0.1, -0.2]$ |
| $\psi_2^*$ | $[-0.3, 0.1, 0.2, -0.1]$ |

The approximating model family (2) used in the experiment is the same as the data-generating process, with all components of parameter vectors $\theta$ and $\psi$ being trainable. The squared MMD discrepancy is defined over Rational Quadratic kernel (Bińkowski et al., 2018)

$$\kappa(y', y'') = \left(1 + \frac{\|y' - y''\|_2^2}{2\,\alpha\,\sigma^2}\right)^{-\alpha} \tag{23}$$

of hyper-parameters $\sigma = 0.04$ and $\alpha = 0.5$ tuned on the validation set.

The model is trained using Adam optimizer (Kingma & Ba, 2014) with parameters $\beta_1 = 0.9$, $\beta_2 = 0.99$. Where not specified, the learning rate is set to $0.05$ and decreased to $0.01$ after 5 epochs. We grouped data points into batches of size 128. Initial values of $\theta$ are independently sampled from the $\mathcal{U}(0.0, 0.1)$ uniform distribution.

## C.2. Additional details on the experiments of Section 6.1

We present here additional figures discussed in Section 6.1. Figure 5 reports the values of the learned parameters $\theta$, while Figure 6 the absolute discrepancy from $\theta^*$. Figure 7 reports the values of the learned parameters $\theta$ when considering a graph of 120 nodes.

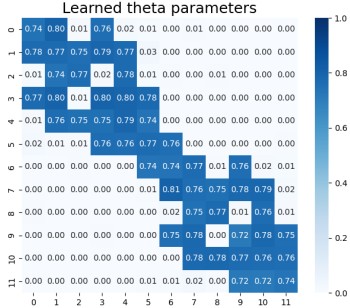

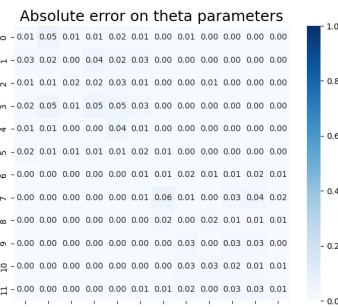

*Figure 5.* The learned parameters for the latent distribution corresponding to the stochastic adjacency matrix.

*Figure 6.* Absolute error made on the parameters of the latent distribution.

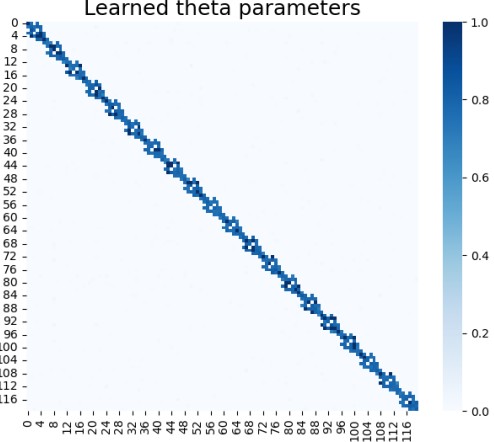

*Figure 7.* Learned $\theta$ parameters for a graph with $\sim 15K$ possible edges.

## C.3. Additional details on the experiments of Section 6.2

We present here additional details discussed in Section 6.2.

**Fixed perturbed $f_\psi$**   Figures in this paragraph correspond to the experiment where the processing function $f_\psi$ is fixed on a perturbed version of $f^*$. Figures $8 - 11$ correspond to runs with increasing perturbation factor $\Psi$.

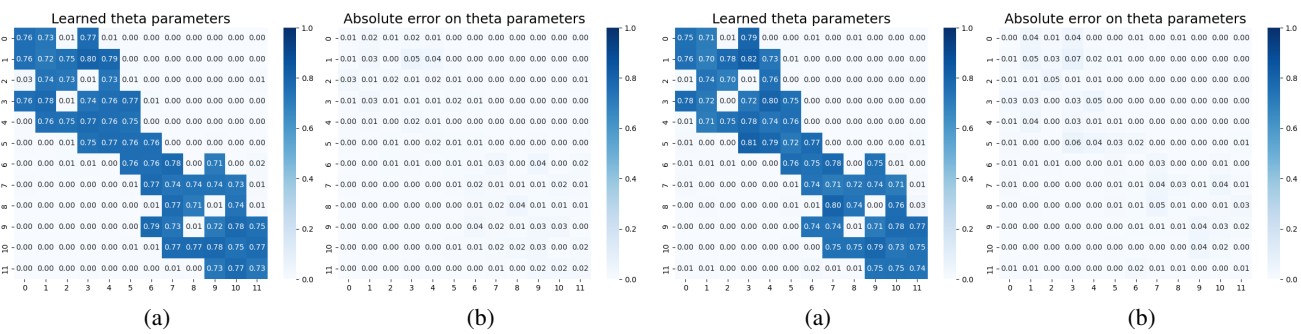

*Figure 8.* Learned $\theta_{ij}$ parameters (a) and Absolute Error (b) for maximum perturbation factor $\Psi$ of 10%.

*Figure 9.* Learned $\theta_{ij}$ parameters (a) and Absolute Error (b) for maximum perturbation factor $\Psi$ of 20%.

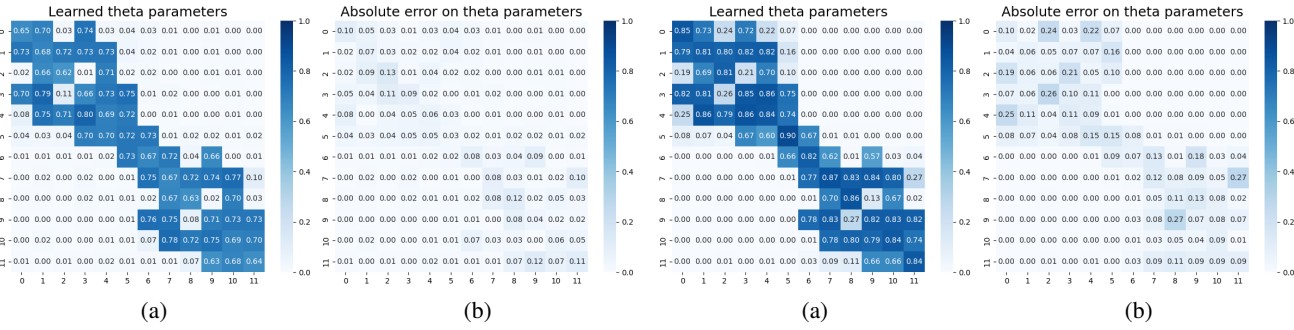

*Figure 10.* Learned $\theta_{ij}$ parameters (a) and Absolute Error (b) for maximum perturbation factor $\Psi$ of 50%.

*Figure 11.* Learned $\theta_{ij}$ parameters (a) and Absolute Error (b) for maximum perturbation factor $\Psi$ of 80%.

**Generic GNN as $f_\psi$** To evaluate our approach in a more realistic setting, we use a generic GNN as $f_\psi$. Specifically, we implement GNNs from (Morris et al., 2019) with varying numbers of layers and layer sizes. It is important to note that the GNN implementation includes self-loops, which prevents the diagonal elements from being correctly learned. However, this does not impede our method from learning the remaining edges accurately.

Table 4 presents the network configurations and whether they successfully converged to the ground truth distribution. Since diagonal elements artificially inflate the MAE for $\theta$, we consider a model to have converged if the final MAE on $\theta$ is less than 0.11.

*Table 4.* Network configurations and corresponding convergence results.

| Layers dimensions | Convergence |
|---|---|
| $[4, 1]$ | ✗ |
| $[4, 1, 1]$ | ✗ |
| $[4, 2, 1]$ | ✓ |
| $[4, 8, 1]$ | ✓ |
| $[4, 8, 2, 1]$ | ✓ |
| $[4, 16, 8, 1]$ | ✓ |
| $[4, 32, 8, 1]$ | ✓ |
| $[4, 64, 8, 1]$ | ✓ |
| $[4, 64, 16, 1]$ | ✓ |
| $[4, 64, 32, 1]$ | ✓ |
| $[4, 8, 8, 4, 1]$ | ✓ |

Most of the models successfully converged, except those with high bias. This demonstrates that our method is effective even beyond Assumption 3.1. In Figure 12 we show the learned parameters of $P_A^\theta$ for a randomly extracted run.

**Misconfigured $P_A^\theta$** Figures 13 and 14 correspond to the experiment where some $\theta_{ij}$ values of $P_A^\theta$ are fixed at incorrect values, while the processing function $f_\psi$ is fixed to the true one. In the community affected by the perturbation, free $\theta_{ij}$ values tend to be sampled more frequently to compensate for the downsampling imposed by the perturbation. Interestingly, all the edges with at least one edge in the second community (75% of the edges) appear unaffected by the perturbation.

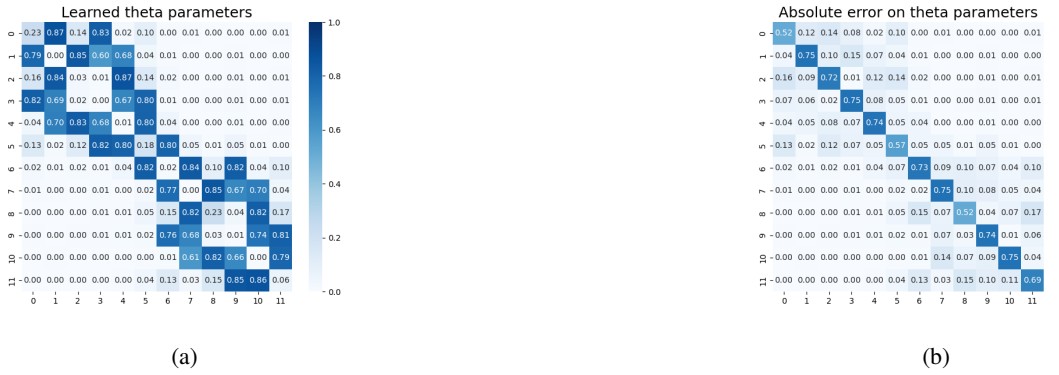

(a)                                                                                          (b)

*Figure 12.* (a) Learned $\theta_{ij}$ parameters when the parametric processing function $f_\psi$ is a generic GNN as presented in (Morris et al., 2019) and (b) Absolute Error made with respect to true parameters $\theta_{ij}^*$. As self-loops are deterministically added by the network, the diagonal elements should not be considered.

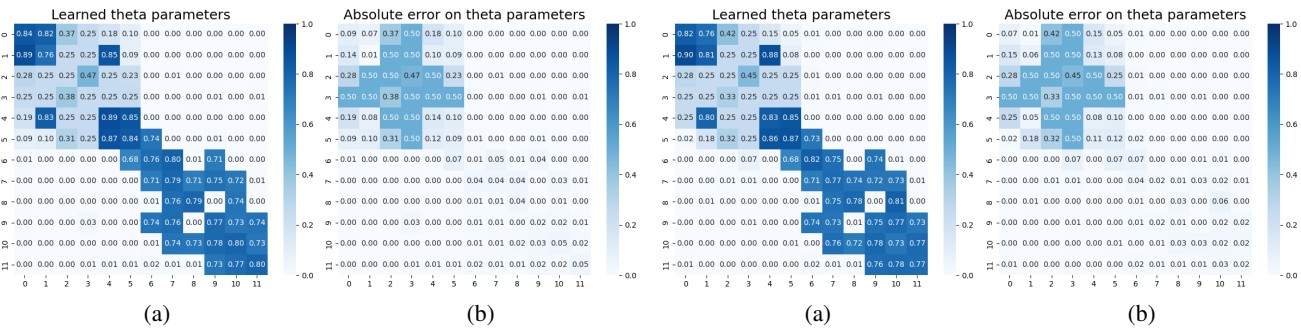

(a)                                    (b)                                    (a)                                    (b)

*Figure 13.* Learned $\theta_{ij}$ parameters (a) and Absolute Error (b) for misconfigured $P_A^\theta$

*Figure 14.* Learned $\theta_{ij}$ parameters (a) and Absolute Error (b) for misconfigured $P_A^\theta$

## C.4. Additional details on the experiments of Section 6.3

All models have been learned using SFE. $\mathcal{L}^{\text{point}}$ rely on the following gradient with respect to $\theta$

$$\nabla_\theta \mathcal{L}^{\text{point}} = \mathbb{E}_{x,y^*}[2(\mathbb{E}_A[\hat{y}] - y^*)\mathbb{E}_A[\hat{y}\nabla_\theta \log P_A^\theta(A)]].$$

For $\mathcal{L}_1^{\text{literature}}$, the gradient is rewritten as

$$\nabla_\theta \mathcal{L}_{1,\ell}^{\text{literature}} = \mathbb{E}_{x,y^*,A}[(\ell(\hat{y},y^*) - b)\nabla_\theta \log P_A^\theta(A)]$$

with $b$ estimating the expected value $\mathbb{E}_{x,y^*,A}[\ell(\hat{y},y^*)]$.

The second family of loss functions $\mathcal{L}_2^{\text{literature}}$ focuses on node-level prediction rewriting $\ell(\hat{y},y^*)$ as the mean $\frac{1}{N}\sum_{i=1}^N \ell(\hat{y}_i,y_i^*)$ over the prediction error at each node $i$. The gradient with respect to $\theta$ is then written as

$$\nabla_\theta \mathcal{L}_{2,\ell}^{\text{literature}} = \mathbb{E}_{x,y^*}\mathbb{E}_{A\sim P_A^\theta}\left[\frac{\sum_i^N (\ell(y_i,y_i^*) - b_i)\nabla_\theta \log(P_A^\theta(A_{i,:}))}{N}\right] \quad (24)$$

where $b_i$ are computed as moving averages of $\ell(y_i,y_i^*)$.

The last family of loss functions (i.e., $\mathcal{L}_{\text{elbo}}^{\text{literature}}$) requires (i) prior distributions $\bar{P}_A(A)$ and $(ii)$ a standard deviations for $P_{y|x^*,A}^\psi(y^*)$ to be set. We consider the following priors and standard deviations, selecting the combination with the lowest validation loss:

(i) For the prior distributions, we assume that each edge is sampled independently according to a Bernoulli distribution. We consider three different prior specifications:

- The first prior is a Bernoulli distribution with parameter $p = 0.01$ for all edges.
- The second prior is a Bernoulli distribution with parameter $p = 0.5$ for all edges.
- The third prior is defined based on the ground truth graph structure: for edges sometimes present in the ground truth structure (i.e., $\theta_{ij}^* \neq 0$), the Bernoulli parameter is $p = 0.75$, while for edges never present in the ground truth structure (i.e., $\theta_{ij}^* = 0$), the parameter is $p = 0.05$.

(ii) the standard deviations considered are: $\{0.001, 0.005, 0.01, 0.05, 0.1, 0.5\}$.

### C.5. Real-world experiment

To demonstrate that our method learns meaningful graph distributions in real-world settings, we train a neural network on air quality data in Beijing (Zheng et al., 2013). The dataset consists of pollutant measurements collected by sensors in Chinese urban areas across several months. We do not use this dataset as a benchmark because the graph structure provided with the data is based on physical distance, and there is no guarantee that it represents the true underlying structure. The neural network we use consists of a GRU unit for processing each time series, followed by a GNN with a learnable graph structure. Figure 15 shows the graph structure learned by our approach, demonstrating its capability to learn meaningful distributions in real-world settings.

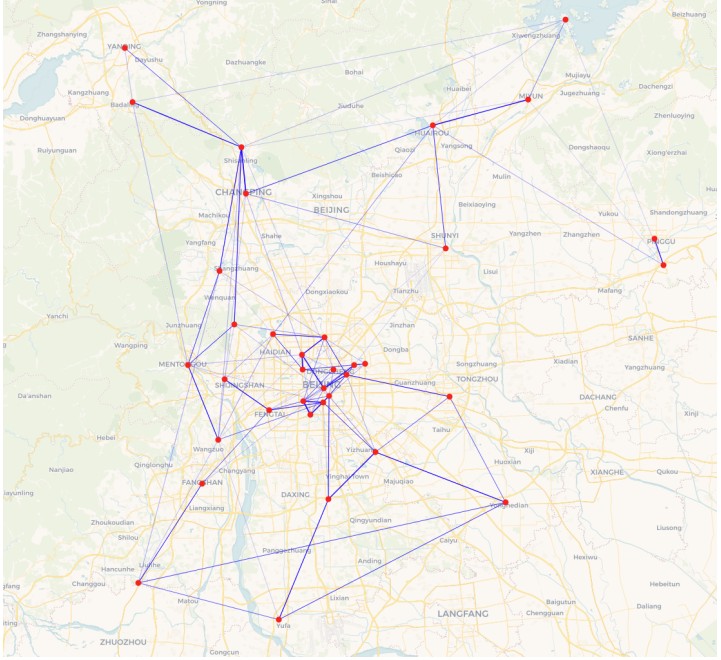

*Figure 15.* Graph structure learned by our approach on Beijing air quality data. The nodes correspond to sensor locations. The thickness of the edges is proportional to the corresponding probability. Map data from OpenStreetMap.

### C.6. Compute resources and open-source software

The paper's experiments were run on a workstation with AMD EPYC 7513 processors and NVIDIA RTX A5000 GPUs; on average, a single model training terminates in a few minutes with a memory usage of about 1GB.

The developed code relies on PyTorch (Paszke et al., 2019) and the following additional open-source libraries: PyTorch Geometric (Fey & Lenssen, 2019), NumPy (Harris et al., 2020) and Matplotlib (Hunter, 2007).

