# OpenReview forum: "Learning Latent Graph Structures and their Uncertainty"
_ICML.cc/2025/Conference — ICML 2025 poster_

### Official Review · Reviewer_qZGr · 2025-02-19

**Overall Recommendation:** 3

**Summary:**

Exploiting the structure of the problem of interest is often key to achieve good generalization with the trained model. For real world applications, it might be known that underlying relational information is shaping the observed data but this latent structure remains often hidden. Some previous works have proposed algorithms learning jointly the topology of the data and the model that leverage it to make predictions in the observational space. Over existing approaches do not quantify how close is the learned latent structure to the true one.
The authors of this submission propose to fill this gap and consider the latent topology as a random object. They provide an algorithm to jointly learn the distribution of the graph latent structure and the one of the observations that relies on it. Their algorithm is supported by a theoretical result showing that minimizing the population loss lead to a well calibrated latent distribution and optimal point prediction.

**Claims And Evidence:**

The theoretical results is validated on a synthetic dataset. The numerical results support the theoretical claims but I think some aspects could be improved:

- I do not totally agree with the authors when they explain that for reall applications, we don't have access to the true distribution of the topology. I think the authors might have considered a case the graph is known and fixed (i.e. the distribution of the latent structure is a Dirac mass) and they could have used their algorithm assuming that the graph is not observed. As the number of observations increased, we expect the learned distribution of the latent graph to converge to this Dirac mass.

- The authors prove Proposition 4.1 by giving a concrete example where calibration is not achieved by minimizing the point-prediction loss. However, this example is very specific and it would have been interesting to run the standard methods (optimizing the point prediction loss) on their synthetic dataset to see if the learned distribution of the latent structure is calibrated or not. This is important since the main contribution claimed by the authors is to propose a way to learn calibrated latent distribution. Therefore it would have been valuable to show that existing methods do not lead to good calibration in most cases.

**Essential References Not Discussed:**

I am not expert in the field and thus other reviewers will have much more relevant comments for this section.

**Experimental Designs Or Analyses:**

Please see my comments in section "Claims And Evidence".

**Methods And Evaluation Criteria:**

Please see my comments in section "Claims And Evidence".

**Other Comments Or Suggestions:**

Here are a few typos:

- Theorem 5.2 contains some typos.

- In Eq.(10), a probability distribution is missing under the last expectation.

- Before section 6.1: "additional specifics are detailed"

- In Section A.3 of the appendix, a $\phi$ is missing when refering to $P^{\theta,\phi}_{y\mid x}$ at several locations.  In the same section at line 664, I think $\psi$ should not have a superscript $*$.

- At line 732, R^N should be \mathbb R^N.

- At line 736: "therfore"

**Other Strengths And Weaknesses:**

None

**Questions For Authors:**

Let me first thank the authors for their work. I really appreciate the effort made to investigate quite deeply both the theoretical (such as in section A.4) and practical (such as section B with the variance reduction method) aspects of their approach.

- Regarding Section A.4, the injectivity hypothesis is discussed for a graph neural network. The analysis shows the assumption is satisfies under mild condition for a very simple GNN. Could you comment on the injectivity assumption for more realistic GNN ? My intuition tells me that the injectivity will be less likely to hold for bigger/deeper network.

- The synthetic dataset considered relies on a probabilistic model to generate graphs which looks a bit adhoc. It would be interesting to consider a more structure and realistic model where probability of connection between two nodes is given by k(x_i,x_j) for some kernel k where x_i and x_j are observed features associated to nodes i and j.

**Relation To Broader Scientific Literature:**

The paper relates to the literature on Graph Structure Learning, and more generally to the problem of identifiability in the statistical literature.

**Theoretical Claims:**

I checked the correctness of the proofs of the theoretical claims.

---

> ### Author Rebuttal · Authors · 2025-04-01
>
> **Claims and Evidence**
>
> **a**.
> First of all, we appreciate your constructive comments. We see your point, but the issue with the proposed approach is not that the graph structure provided in the real-world dataset is not a distribution - we agree that the ground-truth distribution can be a Dirac distribution.
> The problem is that there are no guarantees that the provided graph structure is, or is close to, the underlying graph distribution that generated the data; many times the graph structure provided with the data is built using some heuristics (e.g., node similarity or physical distance between sensors).
> This is one of the reasons why Graph Structure Learning emerged as a field.
> The synthetic dataset - and, ultimately, our theoretical results - serves to overcome this limitation. Nonetheless, we perform the following additional experiment.
>
> **a (additional experiment)**.
> To demonstrate that our method is able to learn sensible graph distributions in real-world settings, we train a neural network on air quality data in Beijing (Zheng, et al. 2013). The neural network consists of a GRU unit to process each time series, followed by a GNN with a learnable graph structure. At the following [link](https://anonymous.4open.science/r/learning_latent_graph_structures_and_their_uncertainty/README.md) we share the graph structure learned by the model with the lowest validation loss. As you can see, our approach learns a reasonable graph.
>
> **b**.
> We agree with you, and indeed we ran that experiment in Section 6.3. Note that the first five rows in Table 2 consider either point-prediction losses or losses from the existing literature.
> As shown, the calibration performance is worse than the proposed method.
>
> **Other Comments Or Suggestions**
>
> Thank you for spotting those typos, we have fixed all of them.
>
>
> **Question For Authors**
>
> **1**.
> At the moment, we have only partial results for generic multi-layer GNNs.
> On the one hand, Corollary A.1 in Appendix A provides a meaningful insight for continuous GNNs: it states that even if a multi-layer GNN fails to be injective for most inputs, injectivity at a single point $\bar x$ is sufficient for our results to apply.
> On the other hand, following the proof of Proposition A.2, if $P_x^*$ is absolutely continuous, then $f(\cdot, A)$ is injective almost surely.
> While we do not have a proof to offer yet, the two comments above are encouraging that a proof might be found.
>
> **2**.
> We considered the multivariate Bernoulli distribution because it is general enough to model all graph distributions over independent edges; moreover, it is widely used in the Graph Structure Learning literature (e.g., Francheschi et al. 2019, Elinas et al. 2020, Sun et al. 2021, Cini et al. 2023).
> We agree that considering other distributions, including input-dependent ones, is relevant. Considering the set of experiments already carried out (7 in the paper and 2 during the rebuttal) and that they already rigorously validate our claims, we leave this extension as future work. Finally we stress that the theoretical results apply to more general distributions. Thank you for your relevant comments.
>
> **References**
>
> - Zheng et al. "U-air: When urban air quality inference meets big data.", 2013.
> - Franceschi et al. "Learning discrete structures for graph neural networks.", 2019.
> - Elinas et al. "Variational inference for graph convolutional networks in the absence of graph data and adversarial settings.", 2020.
> - Sun et al. "Graph structure learning with variational information bottleneck.", 2021.
> - Cini et al. "Sparse graph learning from spatiotemporal time series.", 2023.

---

### Official Review · Reviewer_Y7sL · 2025-03-12

**Overall Recommendation:** 3

**Summary:**

In this paper, the authors investigate the calibration of latent graph structure distributions in the context of graph structure learning. They propose an optimization procedure for a predictive probability model that ensures not only learning the best predictive model but also calibrating the latent distribution of the underlying graph structure. To compute the gradient of the Maximum Mean Discrepancy (MMD) loss, the authors rely on Monte Carlo (MC) sampling and reduce the variance introduced by MC through a control variates approach. Experimental results demonstrate the effectiveness of the proposed method.

**Claims And Evidence:**

The claims are generally well-supported by theoretical analysis and experimental results.

**Essential References Not Discussed:**

NAN

**Experimental Designs Or Analyses:**

The experimental design and analysis are generally sound. Some concerns can be referred to the below 'Weakness'.

**Methods And Evaluation Criteria:**

The proposed method is reasonable.

**Other Comments Or Suggestions:**

NAN

**Other Strengths And Weaknesses:**

•	Strengths:
Overall, I think the problem of calibrating the latent graph structure distribution to be important. The proposed method is grounded in theory and is well organized, and the sampling-based learning approach appears practical. The empirical results are also interesting.


•	Weaknesses:
a.	In Equation (1), y is typically considered a discrete graph label, whereas in Equation (2), \hat{y} (the model output) is usually continuous. The predictive distributions of discrete y and continuous \hat{y} cannot theoretically be the same, so is Assumption 5.1 satisfiable?

b.	The injectivity condition in Theorem 5.2 seems rather strong. For instance, in some graph classification tasks, different subgraph structures A may correspond to the same graph label y.

c.	The designed method does not appear to leverage the unique characteristics of graph data. Moreover, all experiments are conducted on synthetic datasets, which raises concerns about the real-world applicability of this method.

d.	When generating synthetic datasets, assuming each edge is generated independently is overly simplistic. Why not employ more commonly used graph generation models?

e.	The authors do not compare the performance of their method against existing graph structure learning methods.

**Questions For Authors:**

What exactly is x in Equation (1)? Can it be understood as the node feature matrix of the input graph?

**Relation To Broader Scientific Literature:**

NAN

**Theoretical Claims:**

The theoretical claims are clearly presented.

---

> ### Author Rebuttal · Authors · 2025-04-01
>
> **Wa**.
> There might be some misunderstanding here, please allow us to clarify that the assumption is indeed satisfiable.
> Both Equations 1 and 2 can model continuous or discrete outputs, but their respective outputs $y$ and $\hat y$ take values from the *same* set $\mathcal Y$.
> $\Delta$ is a dissimilarity measure between distributions over the same set $\mathcal Y$ for both its arguments (i.e., $P_1,P_2$).
> Assumption 5.1 relates to $\Delta$, and it is known to be satisfied, e.g., by $f$-divergences and some integral probability metrics
>
> **Wb**.
> Please note that our Theorem 4.2 demands only injectivity over a set of inputs with non-zero probability; in the case of Corollary A.1, this reduces to ensuring injectivity at a single point $\bar x$. The assumption relates to the data-generating process and the associated learning problem.
> The particular case of graph classification is relevant due to the discrete nature of $\mathcal Y$. Here, ensuring calibration of the latent variable presents increased complexity, regardless of the chosen loss function.
> According to our results, distributional losses should still be favored over point-prediction losses even in this setting, if calibration of the latent variable is among the goals.
>
>
> **Wc**.
> Please note that Proposition A.2 and Lemma A.3 specifically relate to graph neural networks and we empirically demonstrated that all the developed results are applicable in Graph Structure Learning scenarios.
> Secondly, as commented in the paper, ground-truth knowledge about the latent graph distribution is not available in any real-world datasets we are aware of. Our paper addresses this lack of information in two ways: (i) we derived theoretical guarantees to reduce the need for empirical assessments and (ii) we rigorously validated and tested our approach on synthetic data that provides the required ground truth.
> Nonetheless, to provide evidence of the effectiveness of our method in real-world scenarios, we perform an additional experiment.  Please see our response "**a (additional experiment)**" to Reviewer qZGr.
>
>
> **Wd**.
> Parameterizing the graph structure as a set of Bernoulli distributions allows us to cover *any* distribution over graphs of independent edges. Moreover, this is a common parameterization in the graph structure learning literature (Francheschi et al. 2019, Elinas et al. 2020, Sun et al. 2021, Cini et al. 2023), other than being easier to inspect and visualize. Testing other distributions is indeed possible.
> Furthermore, we stress that the theoretical results are not restricted to Bernoulli distributions.
>
> **We**.
> Please note that we do provide a comparison with existing literature (see Section 6.3 and Table 2). However, as our paper discusses loss functions, we compared the proposed approach with loss functions used in the literature.
>
> **Question**
>
> Yes, $x$ can be understood as the node feature matrix provided as input to the prediction model. It can contain discrete and continuous attributes, as common in GSL setups, but also time series, as in spatiotemporal data analysis.
>
>
> **References**
>
> - Franceschi et al. "Learning discrete structures for graph neural networks.", ICML 2019
> - Elinas et al. "Variational inference for graph convolutional networks in the absence of graph data and adversarial settings.", NeurIPS 2020
> - Sun et al. "Graph structure learning with variational information bottleneck.", AAAI 2021
> - Cini et al. "Sparse graph learning from spatiotemporal time series.", JMLR 2023

---

> > ### Comment · Reviewer_Y7sL · 2025-04-05
> >
> > Thank you for the author's response. Most of my questions are addressed.
> >
> > I will keep my score leaning towards acceptance.

---

### Official Review · Reviewer_vKHP · 2025-03-12

**Overall Recommendation:** 4

**Summary:**

This paper carefully studies the problem of how the optimal latent graph can be learned given observational information from both theoretical and empirical perspectives. It proves that optimizing the usual point prediction does not guarantee calibration of the adjacency matrix distribution. It also provides a loss function that guarantees simultaneous calibration of the latent variable and optimal point predictions. A sampling-based strategy with variance reduction is designed to tractably compute the loss functions. Experimental results on carefully designed synthetic datasets verify the theoretical claims, and also demonstrate the advantages when theoretical assumptions are not fully satisfied.

**Update after rebuttal**

I have read the response from the authors. Since my original recommendation was to accept, I will maintain my recommendation.

**Claims And Evidence:**

The claims are well supported by both theoretical and empirical results.

**Essential References Not Discussed:**

N/A

**Experimental Designs Or Analyses:**

The experimental design is rather comprehensive.

1. The experiments are conducted on synthetic datasets because the ground truth graphs are known and it is easy to manipulate the generation process to create different settings. This is reasonable, because I prefer to consider this paper a theory paper and the experiments are for the proof of concepts.

2. The experimental settings are clearly described and well justified.

3. Under the normal setting, the results show that the MMD loss is suitable for joint learning. The variance reduction component is effective. The method is applicable for large-scale graphs. All these results provide strong support for the theoretical analysis in the main text.

4. Considering that the theoretical analysis relies on Assumption 3.1 which ensures optimal solution, this paper conducts a series of experiments to see what will happen when the assumption is violated. The results show that the proposed method is still effective under different settings.

5. The MMD loss is compared with classical choices of loss functions in GSL. The results show that MMD is advantageous in predicting the target variable and superior in recovering the latent graphs.

**Methods And Evaluation Criteria:**

1. The proposed method is theoretically motivated.

2. Sampling-based estimation of MMD and the control variate for variance reduction are properly derived.

3. The computational complexity is analyzed and it seems acceptable for practice.

4. The method is evaluated comprehensively in experiments. See **Experimental Designs Or Analyses** for detailed comments.

**Other Comments Or Suggestions:**

C1. Even though Assumption 3.1 ensures the existence of optimal parameters, it can be hard to reach in practice. For example, when inferring latent graphs from multivariate time series, prediction error is inevitable, and there could be more than one best graph. The authors can discuss such limitations when applying their theory to real-world applications.

References:

A Graph Dynamics Prior for Relational Inference. In AAAI, 2024.

C2. In Appendix A.2, to argue that minimizing $\mathcal{L}^{point}$ does not guarantee calibration, a counter-example is created based on a specific choice of the loss function, i.e., the MAE. However, in the statement of Proposition 4.1, the authors seem to claim that the conclusion always holds for all choices of $\ell$. Can you make the statement of Proposition 4.1 more precise or provide any proof or thoughts to support that counter-examples can always be constructed?

C3. In Assumption 5.1, can we relax the statement of "$\triangle(P_1,P_2)=0$ if and only if $P_1=P_2$" to "$\triangle(P_1,P_2)=0$ if and only if $P_1=P_2$ almost surely"?

C4. In Appendix B, why is it reasonable to approximate the numerator of Eq. (10) with $\mathbb{E}[L(A)]\mathbb{E}[(\nabla_\theta\log P^\theta(A))^2]$? When will the approximation be accurate?

C5. In Line 687 of the proof of Corollary A.1, should we write $A\not=A'$ to ensure that $\bar{\epsilon}>0$? Besides, since $\delta$ is chosen for $f^*(\cdot,A)$, how can we ensure that $\lVert f^*(\bar{x},A')-f^*(x,A')\rVert<\epsilon$? That is, why does the second inequality in Line 695 hold? I don't fully understand the proof. Can you provide more explanation?

**Other Strengths And Weaknesses:**

Strengths:

S1. This paper explains a long-troubling issue in GSL in theory under some simplified assumptions.

S2. The notations are used accurately, theoretical results are clearly stated and the proofs are concise to read.

Weaknesses:

W1. Figure 2-11 appear in the appendix but are mentioned in the main text. Please consider better organizing the presentation of the experimental results to make the main text more self-contained.

**Questions For Authors:**

N/A

**Relation To Broader Scientific Literature:**

This paper is related to graph structure learning (GSL), latent graph inference, and distribution calibraction.

**Theoretical Claims:**

The theoretical claims on the limitation of point prediction and the optimality of the proposed loss functions are supported by rigorously stated theorems with proofs.

---

> ### Author Rebuttal · Authors · 2025-04-01
>
> **W1**.
> We appreciate your suggestion. If the paper is accepted, we will use part of the camera-ready additional page to improve the presentation.
>
> **C1**.
> Yes, this is a relevant point. This is why we run experiments in controlled settings to test our method beyond those assumptions. In particular, Section 6.1 studies the joint learning problem and shows that the processing function can be approximated well.
> Section 6.2 ("Perturbed $f_{\psi^*}$") tests cases of unsuccessful training, showing that even if the GNN makes prediction errors, the graph structure can be learned with good quality. Section 6.2 ("Generic GNN as $f_{\psi}$") considers a family of GNNs that does not fulfill Assumption 3.1. Also in this setting we were able to learn meaningful distributions.
> Finally, we would like to highlight that point-prediction functions simply fail to calibrate the graph structure even when the GNN is fixed to the true (supposed to be known) processing function.
>
> **C2**.
> Thank you for pointing out this potential source of misunderstanding. Proposition 4.1 states that calibration is not granted for a generic point-prediction loss, not that for any choice it will necessarily fail. In Appendix A.2, we showed that optimizing a commonly adopted metric, the MAE, can be problematic in that context; other specific metrics are not directly investigated in this paper.
> We will revise the paper to prevent any possible confusion.
>
> **C3**.
> Thank you for your relevant question. For the purpose of Theorem 5.2, Assumption 5.1 can be relaxed to hold almost surely with respect to $P_x^*$  - as you suggested. However, we think the current statement of Assumption 5.1 is easier to grasp.
>
> **C4**.
> Thank you for carefully reading the appendices.
> The more the two terms in the expectation are independent the more the approximation is accurate.
> The main reason to accept this approximation is that it enables an easy estimation of the optimal $\beta$.
>
> **C5**.
> Yes, we are considering $A \not = A'$ in that line. We fixed the typo, thank you.
> Regarding the second question, for every $A$ we can find a $\delta_A$ granting $||f^*(\bar{x}, A) -  f^*(x, A)||<\epsilon$; this follows from the continuity of $x\mapsto f^*(x, A)$. As we have finitely many $A$, we take $\delta=\min_{A\in\mathcal A} \delta_A$. Thank you for raising the point, we will clarify it.

---

> > ### Comment · Reviewer_vKHP · 2025-04-02
> >
> > Thank you for resolving all of my concerns. Please revise the paper accordingly. I will keep my overall recommendation.

---

### Official Review · Reviewer_uM1K · 2025-03-20

**Overall Recommendation:** 2

**Summary:**

This paper deals with the problem of learning on graph in a setting where the graph (Adj. matrix: A) is unobserved and is to be estimated from the training data (node features: x, node labels/targets: y). It theoretically shows that the optimal point estimate does not guarantee the calibration of the latent graph structure's distribution. However, if a certain type of loss between the ground truth and model's prediction distribution is zero, then that both achieves the optimal point estimate of the targets and recovers the true distribution of the graph structure under restrictive assumptions. A variance reduction strategy using control variates is adapted for the specific setting. Numerical experiments on small scale graphs are conducted to verify the theory presented.

Overall, the theoretical contribution of this work is inadequate and the empirical validation is rather weak (detailed comments/questions/suggestions in later sections).

**Update:** The authors' rebuttal helped in partially addressing some of my concerns, particularly the comparison with one existing baseline on a synthetic dataset shows that the proposed method can generate graphs with similar statistics. However, I still think that **a)** the novel theoretical contribution (the proofs are extremely simple, MMD is an existing loss function, and control variate is another widespread technique for variance reduction) is limited primarily due to unrealistic and restrictive modelling assumptions, **b)** the experiments seriously lack investigation of performance and comparison with existing techniques with various parameterizations of the graph generative models. I am raising my score to 2 to acknowledge the authors' effort during the rebuttal.

**Claims And Evidence:**

1) Optimal point estimate of the targets does not lead to calibration of  graph distribution: Proposition 4.1 shows that.

2) MMD minimization between the pushforward measures of targets leads to optimal point estimate of the targets and calibration of distribution of A: Theorem 5.2 shows that. (I have some questions about the usefulness of this, which are asked in 'questions to authors' section below)

**Essential References Not Discussed:**

N/A

**Experimental Designs Or Analyses:**

The experiments numerically verify the theory presented (my questions are written below)

**Methods And Evaluation Criteria:**

Results in Table 2 verifies the theory presented numerically.

**Other Comments Or Suggestions:**

N/A

**Other Strengths And Weaknesses:**

Strengths:

1) The paper is moderately well written and easy to read.

Weaknesses:

1) Relevance to GNN community: The main theoretical results presented in this paper are only marginally relevant to advancing knowledge in the area of learning on graphs. Specifically, the theoretical results are valid for any latent variable model which can be characterized by eq 1. The stated assumptions are not specific to graph domain in any discernable way as well. It is not clear how the results of this paper can be helpful for any GNN researcher or practitioner in any way.

2) Restrictive modelling assumptions: The model in eq. 1 is rather limited. First, this idea that $y= f(x, A)$ is a deterministic function of $x$ and $A$ limits its applicability to classification problems, where the label is a K-ary variable, sampled from $p(y|x, A)$. Similarly, if the targets are noisy in a regression setting, then a deterministic mapping is inadequate.

3) Impractical modelling assumptions: The model in eq. 1 considers $x$ observed and factorize the joint distribution as follows:
$p(y, A|x) = p(A) \times Indicator_fun(y=f(A, x))$. In other words, eq. 1 does not consider any dependency between A and x. But, this is typically not the case in real world graph datasets and for problems of practical interest.  For example, Kalofolias 2016 (https://arxiv.org/pdf/1601.02513) considers learning a graph from smooth signals. Intuitively, if the node features are similar for a node pair $(i,j)$, then $A_{i,j}$ should be higher and vice versa, this type of approach models the dependency between $x$ and $A$ explicitly. Even for the datasets, used extensively for evaluation of GNNs, there is dependency between the graph structure and node features. For example, on homophilic graphs (e.g. Cora, having stochastic block model structure approximately), the node feature similarity is heavily correlated with graph connectivity. It is not clear whether the results in this work can address such setting.

4) Other assumptions: How does the injectivity assumption work for any GNN with more than one layers? Otherwise, theorem 5.2 is of little practical significance.

5) Other comments about theoretical results: The 'proof' of Proposition 4.1 only requires the understanding that two distributions can differ even if a certain moment matches. Given the trivial nature of this results, the choice of stating it as a formal proposition is questionable.
Moreover, the algorithm does not guarantee that we can obtain $\phi^*$, so how the requirement of knowledge of the 'true' GNN in Theorem 5.2 is justified?

5) While the authors argue in favor of not including any real data experiments and existing baselines, it is not at all convincing. For example, if the 'true' distribution that generates the graph is unknown, one could still run the proposed method with some chosen parameterization (to capture some useful inductive bias) to perform an estimation of $P(A)$, and then use the learned model for sampling graphs and then check whether graph statistics match with the 'true' distribution. This avenue gives an indirect route to assess calibration. Bayesian GNNs (Zhang et al., 2019) and Variational GNN (Elinas et al., 2020) could serve as baselines with minimal modifications in that setting.

6) Discussion of related work is far from being accurate. For example, the authors write "Some approaches from
the literature model the latent graph structure as stochastic (Kipf et al., 2018; Franceschi et al., 2019; Elinas et al.,
2020; Shang et al., 2021; Cini et al., 2023), mainly as a
way to enforce sparsity of the adjacency matrix."  Kipf et al., 2018 propose GCN on a fixed graph, they do NOT 'model the latent graph structure as stochastic '.

**Questions For Authors:**

Please address the issues raised in the previous sections.

**Relation To Broader Scientific Literature:**

While the results are correct, their impact in advancing the field of learning on graphs is minimal in my view, because of impractical and restrictive modelling assumptions and lack of real-data experiments (check the weakness section for detailed comments and questions).

The discussion of related work is confusing and inaccurate at places (check the weakness section for details).

**Theoretical Claims:**

Please see "Claims And Evidence" section above.

---

> ### Author Rebuttal · Authors · 2025-04-01
>
> **W1**.
> Part of our results can indeed be applied to more general latent variable models, and we view this broader applicability as a strength rather than a weakness. As shown in the Experiments Section, our theoretical results can be successfully applied to Graph Structure Learning problems, making them relevant to the GNN community. Furthermore, some results are specific to GNNs (e.g., Proposition A.2).
>
> **W2 W3, W4 and W5**.
> Our paper highlights the critical role of the loss function to effectively achieve calibration; this specific formulation and set of assumptions serve this goal.
> Some assumptions can be relaxed, broadening the applicability of our findings. However, we believe that our developments lay a solid foundation enabling further developments.
> We expect - and have validated with different experiments (refer to Sections 6.1 and 6.2) - that the theory remains valid beyond certain assumptions.
> In the following points, we provide additional evidence to support that our results can be generalized to address your concerns further:
>
> - **W2)**
> We performed an additional experiment where $p(y|x,A)$ is a continuous distribution both in the system model (Eq. 1) and the approximating model (Eq. 2). Specifically, we adapt the data-generating process described in Section 6 to include uniform noise $\eta \sim \mathcal{U}(-\Psi^*, \Psi^*)$) added to each of the components of the output $y$.
> Accordingly, the approximating model now includes a learnable stochastic variable ($y=f_\psi(x,A) + \epsilon$ with $\epsilon \sim\mathcal{U}(-\Psi, \Psi)$ and $\Psi$ learnable).
> The results (reported below for different values of $\Psi^*$) demonstrate that our approach is able to approximate the real distribution even if the processing function is not deterministic.
> | Max pert. $\Psi^*$ | MAE on $\theta$ | Max AE on $\theta$ |
> | --- | --- | --- |
> | 0    | 0.009 $\pm$ 0.001 | 0.06 $\pm$ 0.01 |
> | 0.1  | 0.008 $\pm$ 0.001 | 0.05 $\pm$ 0.01 |
> | 0.2  | 0.012 $\pm$ 0.001 | 0.06 $\pm$ 0.01 |
> | 0.3  | 0.019 $\pm$ 0.003 | 0.09 $\pm$ 0.01 |
> | 0.4  | 0.032 $\pm$ 0.003 | 0.13 $\pm$ 0.01 |
>
>
> - **W3)**
> It is indeed possible to parametrize the graph distribution making it input dependent.
> One approach involves parameterizing $\theta$, the parameters of $P_A^\theta$, as a function of the input $x$, such as $\theta = g_{\theta'}(x)$, where $\theta'$ denotes an additional set of free parameters. Regarding the theoretical results, Theorem 4.2 can be extended, e.g., to piecewise constant mappings $x \mapsto p(A|x)$; we conjecture that a proof could be established for $p(A|x)$ smoothly varying with respect to $x$.
>
>
> - **W4)**
> Please see our response to Reviewer qZGr's Question 1.
>
>
> - **W5)**
> In the paper, we conducted various experiments to study this theorem's hypothesis, specifically (a) solving the joint learning problem (Section 6.1), (b) setting the processing function to a different, incorrect function (Section 6.2 "Perturbed $f_{\psi^*}$), and (c) using a generic GNN from a different class compared to the true one (Section 6.2 "Generic GNN as $f_{\psi}$"). The empirical evidence gathered demonstrates that, even in these cases, we can still appropriately learn the latent distribution.
> Please note that we have also demonstrated that point-prediction loss functions often fail to calibrate the latent distribution, even when the true processing function is used (i.e., $\psi = \psi*$).
>
>
> **W6**.
> We are not sure we have fully understood your proposed solution.
> In real-world applications, $P^*(A)$ is unknown and no samples from it are observed. How do you suggest estimating it? Assuming a priori that one model is better than another and, therefore, that it could serve as a ground truth seems inappropriate to us. If our response does not address your comment, could you elaborate in more detail?
> Nonetheless, to provide additional evidence that our method learns sensible graph distributions in real-world settings, we run an extra experiment. Please refer to our response "**a (additional experiment)**" to Reviewer qZGr.
>
>
>  **W7**.
> Respectfully, we disagree on this point. Kipf et al. (2018) do model the latent relationships as stochastic. Citing from their paper, they use an "encoder that predicts a probability distribution $q_\phi(z|x)$ over the latent interactions given input trajectories", more in detail the encoder "returns a factorized distribution of $z_{ij}$, where $z_{ij}$ is a discrete categorical variable representing the edge type between object $v_i$ and $v_j$". In Section 3.2 they further show how they used the Gumbel-Softmax trick (Maddison et al., 2017) to sample from this discrete latent random variable.
>
> **References**
>
> - Kipf et al. "Neural relational inference for interacting systems.", ICML 2018
> - Maddison et al.. "The concrete distribution: A continuous relaxation of discrete random variables.", ICLR 2017

---

> > ### Comment · Reviewer_uM1K · 2025-04-03
> >
> > I thank the authors for their rebuttal, it has helped to clarify some aspects of the work better.
> >
> > I apologize for the mistake I made in my comment on the discussion of (Kipf et al. 2018). I confused this work with the popular (Kipf et al. 2017) GCN paper. Now that I have checked the paper that the authors cited there, it is appropriate to be cited.
> >
> > However, my question on comparison to existing baselines remain, please allow me to explain in detail how such a comparison can be done on a **synthetic** dataset and correct me if I am wrong.
> >
> > For the training of this approach, one needs a dataset of features and labels. Now we use the same dataset to train the model in  Elinas et al. (https://arxiv.org/pdf/1906.01852 , note that this approach does not need an observed graph for training). After training, we will have the approximation of the graph posterior $p(A|X, Y)$. One can sample several graphs from this distribution and compute various graph statistics. Similarly, from the proposed approach, we can sample several graphs (using the learned $\theta$) after the training is complete and calculate the same statistics.
> >
> > Now, we would like to know, in which case, these graph statistics are closer to the same statistics of the 'true' graph(s) used to create the dataset. If the empirical calibration offered by this proposed approach is indeed better, it should be reflected in that comparison.  In addition, one could compare the performance of these two methods in terms of estimating $y$.
> >
> >
> > Without these sort of results, my current impression is that a) yes, the numerical experiments verify the theory presented, b) however, there are existing approaches (Elinas et al. ), which operate in the same setting (i.e. estimate the graph distribution and labels together) , c) and, no comparison either in terms of graph estimation and/or label estimation is presented.
> >
> > Note: Elinas et al. considers a classification setting, but, reading their methodology, extending to the regression setting should not be a major problem.

---

> > > ### Author Response · Authors · 2025-04-07
> > >
> > > We appreciate your clarifications, which help us understand and address your comments.
> > >
> > > What you requested has been partly implemented already, and we are happy to integrate it.
> > > We performed two additional experiments following your suggestion. The first one compares our methods and implemented baselines (Table 2) with an approach following Elinas et al. (2020).
> > > The second experiment addresses your comment on calibration assessment.
> > > We anticipate that (1) results are consistent with our paper's main argument and (2) considered distributional losses maintain improved calibration performance also according to the approach you proposed.
> > >
> > > More in detail:
> > >
> > > **Experiment 1.** We trained a model optimizing the ELBO loss in Elinas et al. (2020), adapting it for the synthetic regression task as suggested.
> > > The loss is  $\mathcal{L}^{elbo}(\theta,\psi)$ $= - E_{x^*\sim P_x^*}[E_{A \sim P_A^\theta(A)}[ \text{ log }(P^\psi_{y|x^* A}(y^*))]] + KL[P_A^\theta(A)|| \bar{P}_A(A)]$, with prior distribution $\bar{P}_A(A)$ and
> > >
> > > $P^\psi_{y|x^* A}(y^*)$ a Gaussian distribution in this case.
> > > We swept through different hyperparameters and choices of the prior $\bar P(A)$, and trained all other parameters $\theta$ and $\psi$ as per Table 2. The overall best results for $\mathcal L^{elbo}$ are as follows:
> > >
> > > | Loss | MAE on $\theta$ | MAE on $y$ | MSE on $y$ |
> > > | --- | --- | --- | --- |
> > > | $\mathcal L^{elbo}$  |  0.082 $\pm$ 0.001 | 0.31 $\pm$ 0.01  |  0.19 $\pm$ 0.02 |
> > > | $\mathcal L^{dist}_{\Delta:\text{MMD}}$  |  0.010 $\pm$ 0.002  |  0.269 $\pm$ 0.001  |  0.159 $\pm$ 0.001 |
> > >
> > > values for $\mathcal L^{dist}_{\Delta:\text{MMD}}$ are reported from Table 2.
> > >
> > > While different hyperparameters yield different tradeoffs between prediction accuracy and model calibration, we observed that the performance is not on par with that achieved using distributional losses.
> > >
> > > **Experiment 2.**
> > > To address your request regarding the empirical assessment of model calibration, we compared the learned distributions with the ground-truth one $P_A^*$ with different graph statistics. We considered the input-output pairs from the dataset of Section 6 and the following graph statistics:
> > >
> > > | Graph statistic | $P_A^{point}$ | $P_A^{elbo}$ | $P_A^{dist}$ | $P_A^*$ |
> > > | --- | --- | --- | --- | --- |
> > > | Average node degree  | $3.28 \pm 0.06$  | $4.12 \pm 0.01$  | $3.166 \pm 0.008$  |  $3.120 \pm 0.003$ |
> > > | Average number edges  | $39.3 \pm 0.7$  | $49.4 \pm 0.01$  | $37.98 \pm 0.09$  |  $37.44 \pm 0.03$ |
> > > | Number of triangles   | $9.2 \pm 0.5$  | $8.57 \pm 0.08$  | $6.92 \pm 0.04$  |  $6.58 \pm 0.05$ |
> > >
> > > In the table:
> > > - $P_A^{dist}$ is learned by optimizing $\mathcal{L}_{\Delta:\text{MMD}}^{dist}$,
> > >
> > > - $P_A^{point}$ by optimizing $\mathcal{L}^{point}_{\ell:\text{MSE}}$,
> > >
> > > - $P_A^{elbo}$ by optimizing $\mathcal{L}^{elbo}$ from Experiment 1.
> > >
> > > Graph statistics were computed from a sample of 500 graphs and repeating the training 3 times. The reported results are consistent with our paper's findings.
> > >
> > > Regarding the specific points a), b) and c)
> > >
> > > - **Point a)** Indeed this is one of our contributions.
> > >
> > > - **Point b)** We agree, and we acknowledge it in our paper; please see from line 046. Further, we performed the additional Experiment 1 described above.
> > >
> > > - **Point c) graph estimation** We fulfill this point in two ways. First, we compared the estimated graph distributions in Section 6.3, considering various approaches; specifically, Table 2 considers point-prediction and distributional losses, as well as other losses from the literature. Secondly, we performed the additional two experiments described above, following the reviewer's indications.
> > >
> > > - **Point c) label estimation** We do assess the quality of predictions on the target $y$. Our analyses report the accuracy in estimating the expected value of $E_{y^*\sim P^*_{y|x^*}}[y^*]$ and the median median$(y^*)$; please, refer to MSE and MAE on $y$ in Table 2.
> > >
> > > To conclude, we believe all three points **a)**, **b)** and **c)** to be resolved now.

---

### Decision · Program_Chairs · 2025-05-01

**Decision:**

Accept (poster)

**Comment:**

This paper studies graph structure learning and the associated uncertainty in the learning process, whose goal is to learn a calibrated latent graph distribution to reflect the uncertainty. This paper shows theoretically that, under certain conditions, minimizing output distribution loss yields calibration of latent variables and optimal pointwise prediction. The paper achieves it via MMD minimization and MC sampling is done for optimization.

Main concerns of the paper were mainly about motivation of this paper (no access to the distribution of graphs), theoretical analysis (injectivity, relationship between proposed method and graph structure, independent edge generation, certain proof techniques), application to real-world scenarios, discussions on existing works, and additional experiments about graph/label estimation. I've went through the manuscript, reviewers' concerns, and author-reviewer discussion. Overall, the concerns are adequately addressed with enough theoretical and empirical evidence in my opinion. The topic (calibrated GSL) is of importance to many real-world applications. I recommend acceptance.